# Stochastic Online Greedy Learning with Semi-bandit Feedbacks

**Tian Lin**
Tsinghua University
Beijing, China
lintian06@gmail.com

**Jian Li**
Tsinghua University
Beijing, China
lapordge@gmail.com

**Wei Chen**
Microsoft Research
Beijing, China
weic@microsoft.com

## Abstract

The greedy algorithm is extensively studied in the field of combinatorial optimization for decades. In this paper, we address the online learning problem when the input to the greedy algorithm is stochastic with unknown parameters that have to be learned over time. We first propose the greedy regret and $\epsilon$-quasi greedy regret as learning metrics comparing with the performance of offline greedy algorithm. We then propose two online greedy learning algorithms with semi-bandit feedbacks, which use multi-armed bandit and pure exploration bandit policies at each level of greedy learning, one for each of the regret metrics respectively. Both algorithms achieve $O(\log T)$ problem-dependent regret bound ($T$ being the time horizon) for a general class of combinatorial structures and reward functions that allow greedy solutions. We further show that the bound is tight in $T$ and other problem instance parameters.

## 1 Introduction

The greedy algorithm is simple and easy-to-implement, and can be applied to solve a wide range of complex optimization problems, either with exact solutions (e.g. minimum spanning tree [19, 25]) or approximate solutions (e.g. maximum coverage [11] or influence maximization [17]). Moreover, for many practical problems, the greedy algorithm often serves as the first heuristic of choice and performs well in practice even when it does not provide a theoretical guarantee.

The classical greedy algorithm assumes that a certain *reward function* is given, and it constructs the solution iteratively. In *each phase*, it searches for a local optimal element to maximize the marginal gain of reward, and add it to the solution. We refer to this case as the *offline greedy* algorithm with a given reward function, and the corresponding problem the *offline problems*. The phase-by-phase process of the greedy algorithm naturally forms a *decision sequence* to illustrate the decision flow in finding the solution, which is named as the *greedy sequence*. We characterize the decision class as an *accessible set system*, a general combinatorial structure encompassing many interesting problems.

In many real applications, however, the reward function is *stochastic* and is not known in advance, and the reward is only instantiated based on the unknown distribution after the greedy sequence is selected. For example, in the influence maximization problem [17], social influence are propagated in a social network from the selected seed nodes following a stochastic model with unknown parameters, and one wants to find the optimal seed set of size $k$ that generates the largest influence spread, which is the expected number of nodes influenced in a cascade. In this case, the reward of seed selection is only instantiated after the seed selection, and is only one of the random outcomes. Therefore, when the stochastic reward function is unknown, we aim at maximizing the expected reward overtime while gradually learning the key parameters of the expected reward functions. This falls in the domain of *online learning*, and we refer the *online algorithm* as the strategy of the *player*, who makes sequential decisions, interacts with the environment, obtains feedbacks, and accumulates

her reward. For *online greedy algorithms* in particular, at each time step the player selects and plays a candidate decision sequence while the environment instantiates the reward function, and then the player collects the values of instantiated function at every phase of the decision sequence as the feedbacks (thus the name of *semi-bandit feedbacks* [2]), and takes the value of the final phase as the reward cumulated in this step.

The typical objective for an online algorithm is to make sequential decisions against the optimal solution in the offline problem where the reward function is known a priori. For online greedy algorithms, instead, we compare it with the solution of the offline greedy algorithm, and minimize their gap of the cumulative reward over time, termed as the *greedy regret*. Furthermore, in some problems such as influence maximization, the reward function is estimated with error even for the offline problem [17] and thus the greedily selected element at each phase may contain some $\epsilon$ error. We call such greedy sequence as $\epsilon$-*quasi greedy sequence*. To accommodate these cases, we also define the metric of $\epsilon$-*quasi greedy regret*, which compares the online solution against the minimum offline solution from all $\epsilon$-quasi greedy sequences.

In this paper, we propose two online greedy algorithms targeted at two regret metrics respectively. The first algorithm OG-UCB uses the stochastic multi-armed bandit (MAB) [22, 8], in particular the well-known UCB policy [3] as the building block to minimize the greedy regret. We apply the UCB policy to every phase by associating the confidence bound to each arm, and then choose the arm having the highest upper confidence bound greedily in the process of decision. For the second scenario where we allow tolerating $\epsilon$-error for each phase, we propose a *first-explore-then-exploit* algorithm OG-LUCB to minimize the $\epsilon$-quasi greedy regret. For every phase in the greedy process, OG-LUCB applies the LUCB policy [16, 9] which depends on the upper and lower confidence bound to eliminate arms. It first explores each arm until the lower bound of one arm is higher than the upper bound of any other arm within an $\epsilon$-error, then the stage of current phase is switched to exploit that best arm, and continues to the next phase. Both OG-UCB and OG-LUCB achieve the problem-dependent $O(\log T)$ bound in terms of the respective regret metrics, where the coefficients in front of $T$ depends on direct elements along the greedy sequence (a.k.a., its *decision frontier*) corresponding to the instance of learning problem. The two algorithms have complementary advantages: when we really target at greedy regret (setting $\epsilon$ to 0 for OG-LUCB), OG-UCB has a slightly better regret guarantee and does not need an artificial switch between exploration and exploitation; when we are satisfied with $\epsilon$-quasi greedy regret, OG-LUCB works but OG-UCB cannot be adapted for this case and may suffer a larger regret. We also show a problem instance in this paper, where the upper bound is tight to the lower bound in $T$ and other problem parameters.

We further show our algorithms can be easily extended to the knapsack problem, and applied to the stochastic online maximization for consistent functions and submodular functions, etc., in the supplementary material.

To summarize, our contributions include the following: (a) To the best of our knowledge, we are the first to propose the framework using the greedy regret and $\epsilon$-quasi greedy regret to characterize the online performance of the stochastic greedy algorithm for different scenarios, and it works for a wide class of accessible set systems and general reward functions; (b) We propose Algorithms OG-UCB and OG-LUCB that achieve the problem-dependent $O(\log T)$ regret bound; and (c) We also show that the upper bound matches with the lower bound (up to a constant factor).

Due to the space constraint, the analysis of algorithms, applications and empirical evaluation of the lower bound are moved to the supplementary material.

**Related Work.** The multi-armed bandit (MAB) problem for both stochastic and adversarial settings [22, 4, 6] has been widely studied for decades. Most work focus on minimizing the cumulative regret over time [3, 14], or identifying the optimal solution in terms of pure exploration bandits [1, 16, 7]. Among those work, there is one line of research that generalizes MAB to combinatorial learning problems [8, 13, 2, 10, 21, 23, 9]. Our paper belongs to this line considering stochastic learning with semi-bandit feedbacks, while we focus on the greedy algorithm, the structure and its performance measure, which have not been addressed.

The classical greedy algorithms in the offline setting are studied in many applications [19, 25, 11, 5], and there is a line of work [15, 18] focusing on characterizing the greedy structure for solutions. We adopt their characterizations of accessible set systems to the online setting of the greedy learning. There is also a branch of work using the greedy algorithm to solve online learning problem, while

they require the knowledge of the exact form of reward function, restricting to special functions such as linear [2, 20] and submodular rewards [26, 12]. Our work does not assume the exact form, and it covers a much larger class of combinatorial structures and reward functions.

## 2   Preliminaries

Online combinatorial learning problem can be formulated as a repeated game between *the environment* and *the player* under stochastic multi-armed bandit framework.

Let $E = \{e_1, e_2, \ldots, e_n\}$ be a finite *ground set* of size $n$, and $\mathcal{F}$ be a collection of subsets of $E$. We consider the *accessible set system* $(E, \mathcal{F})$ satisfying the following two axioms: (1) $\emptyset \in \mathcal{F}$; (2) If $S \in \mathcal{F}$ and $S \neq \emptyset$, then there exists some $e$ in $E$, s.t., $S \setminus \{e\} \in \mathcal{F}$. We define any set $S \subseteq E$ as a *feasible set* if $S \in \mathcal{F}$. For any $S \in \mathcal{F}$, its *accessible set* is defined as $\mathcal{N}(S) := \{e \in E \setminus S : S \cup \{e\} \in \mathcal{F}\}$. We say feasible set $S$ is *maximal* if $\mathcal{N}(S) = \emptyset$. Define the *largest length* of any feasible set as $m := \max_{S \in \mathcal{F}} |S|$ ($m \leq n$), and the *largest width* of any feasible set as $W := \max_{S \in \mathcal{F}} |\mathcal{N}(S)|$ ($W \leq n$). We say that such an *accessible set system* $(E, \mathcal{F})$ is the *decision class* of the player. In the class of combinatorial learning problems, the size of $\mathcal{F}$ is usually very large (e.g., exponential in $m$, $W$ and $n$).

Beginning with an empty set, the *accessible set system* $(E, \mathcal{F})$ ensures that any feasible set $S$ can be acquired by adding elements one by one in some order (cf. Lemma A.1 in the supplementary material for more details), which naturally forms the decision process of the player. For convenience, we say the player can choose a *decision sequence*, defined as an ordered feasible sets $\sigma := \langle S_0, S_1, \ldots, S_k \rangle \in \mathcal{F}^{k+1}$ satisfying that $\emptyset = S_0 \subset S_1 \subset \cdots \subset S_k$ and for any $i = 1, 2, \ldots, k$, $S_i = S_{i-1} \cup \{s_i\}$ where $s_i \in \mathcal{N}(S_{i-1})$. Besides, define decision sequence $\sigma$ as *maximal* if and only if $S_k$ is maximal.

Let $\Omega$ be an arbitrary set. The environment draws i.i.d. samples from $\Omega$ as $\omega_1, \omega_2, \ldots$, at *each time* $t = 1, 2, \ldots$, by following a predetermined but unknown distribution. Consider reward function $f : \mathcal{F} \times \Omega \to \mathbb{R}$ that is bounded, and it is non-decreasing[1] in the first parameter, while the exact form of function is agnostic to the player. We use a shorthand $f_t(S) := f(S, \omega_t)$ to denote the reward for any given $S$ at time $t$, and denote the expected reward as $\overline{f}(S) := \mathbb{E}_{\omega_1}[f_1(S)]$, where the expectation $\mathbb{E}_{\omega_t}$ is taken from the randomness of the environment at time $t$. For ease of presentation, we assume that the reward function for any time $t$ is normalized with arbitrary alignment as follows: (1) $f_t(\emptyset) = L$ (for any constant $L \geq 0$); (2) for any $S \in \mathcal{F}, e \in \mathcal{N}(S), f_t(S \cup \{e\}) - f_t(S) \in [0, 1]$. Therefore, reward function $f(\cdot, \cdot)$ is implicitly bounded within $[L, L + m]$.

We extend the concept of arms in MAB, and introduce notation $a := e|S$ to define *an arm*, representing the selected element $e$ based on the prefix $S$, where $S$ is a feasible set and $e \in \mathcal{N}(S)$; and define $\mathcal{A} := \{e|S : \forall S \in \mathcal{F}, \forall e \in \mathcal{N}(S)\}$ as the *arm space*. Then, we can define the *marginal reward* for function $f_t$ as $f_t(e|S) := f_t(S \cup \{e\}) - f_t(S)$, and the *expected marginal reward* for $\overline{f}$ as $\overline{f}(e|S) := \overline{f}(S \cup \{e\}) - \overline{f}(S)$. Notice that the use of arms characterizes the marginal reward, and also indicates that it is related to the player's previous decision.

### 2.1   The Offline Problem and The Offline Greedy Algorithm

In the *offline problem*, we assume that $\overline{f}$ is provided as a value oracle. Therefore, the objective is to find the optimal solution $S^* = \arg\max_{S \in \mathcal{F}} \overline{f}(S)$, which only depends on the player's decision. When the optimal solution is computationally hard to obtain, usually we are interested in finding a feasible set $S^+ \in \mathcal{F}$ such that $\overline{f}(S^+) \geq \alpha \overline{f}(S^*)$ where $\alpha \in (0, 1]$, then $S^+$ is called an $\alpha$-*approximation* solution. That is a typical case where the greedy algorithm comes into play.

The *offline greedy* algorithm is a local search algorithm that refines the solution *phase by phase*. It goes as follows: (a) Let $G_0 = \emptyset$; (b) For each phase $k = 0, 1, \ldots$, find $g_{k+1} = \arg\max_{e \in \mathcal{N}(G_k)} \overline{f}(e|G_k)$, and let $G_{k+1} = G_k \cup \{g_{k+1}\}$; (c) The above process ends when $\mathcal{N}(G_{k+1}) = \emptyset$ ($G_{k+1}$ is maximal). We define the maximal decision sequence $\sigma^{\mathsf{G}} := \langle G_0, G_1, \ldots, G_{m^{\mathsf{G}}} \rangle$ ($m^{\mathsf{G}}$ is its length) found by the offline greedy as the *greedy sequence*. For simplicity, we assume that it is unique.

One important feature is that the greedy algorithm uses a polynomial number of calls ($\mathrm{poly}(m, W, n)$) to the offline oracle, even though the size of $\mathcal{F}$ or $\mathcal{A}$ may be exponentially large.

In some cases such as the offline influence maximization problem [17], the value of $\overline{f}(\cdot)$ can only be accessed with some error or estimated approximately. Sometimes, even though $\overline{f}(\cdot)$ can be computed exactly, we may only need an approximate maximizer in each greedy phase in favor of computational efficiency (e.g., efficient submodular maximization [24]). To capture such scenarios, we say a maximal decision sequence $\sigma = \langle S_0, S_1, \ldots, S_{m'} \rangle$ is an $\epsilon$-*quasi greedy sequence* ($\epsilon \geq 0$), if the greedy decision can tolerate $\epsilon$ error every phase, i.e., for each $k = 0, 1, \ldots, m' - 1$ and $S_{k+1} = S_k \cup \{s_{k+1}\}$, $\overline{f}(s_{k+1}|S_k) \geq \max_{s \in \mathcal{N}(S_k)} \overline{f}(s|S_k) - \epsilon$. Notice that there could be many $\epsilon$-quasi greedy sequences, and we denote $\sigma^{\mathsf{Q}} := \langle Q_0, Q_1, \ldots, Q_{m^{\mathsf{Q}}} \rangle$ ($m^{\mathsf{Q}}$ is its length) as the one with the minimum reward, that is $\overline{f}(Q_{m^{\mathsf{Q}}})$ is minimized over all $\epsilon$-quasi greedy sequences.

## 2.2 The Online Problem

In the online case, in constrast $\overline{f}$ is not provided. The player can only access one of functions $f_1, f_2, \ldots$, generated by the environment, for each time step during a repeated game.

For each time $t$, the game proceeds in the following three steps: (1) The environment draws i.i.d. sample $\omega_t \in \Omega$ from its predetermined distribution without revealing it; (2) the player may, based on her previous knowledge, select a decision sequence $\sigma^t = \langle S_0, S_1, \ldots, S_{m^t} \rangle$, which reflects the process of her decision phase by phase; (3) then, the player plays $\sigma^t$ and gains reward $f_t(S_{m^t})$, while observes intermediate feedbacks $f_t(S_0), f_t(S_1), \ldots, f_t(S_{m^t})$ to update her knowledge. We refer such feedbacks as *semi-bandit feedbacks* in the decision order.

For any time $t = 1, 2, \ldots$, denote $\sigma^t = \langle S_0^t, S_1^t, \ldots, S_{m^t}^t \rangle$ and $S^t := S_{m^t}^t$. The player is to make sequential decisions, and the classical objective is to minimize the cumulative gap of rewards against the optimal solution [3] or the approximation solution [10]. For example, when the optimal solution $S^* = \arg \max_{S \in \mathcal{F}} \mathbb{E}[f_1(S)]$ can be solved in the offline problem, we minimize the expected *cumulative regret* $R(T) := T \cdot \mathbb{E}[f_1(S^*)] - \sum_{t=1}^{T} \mathbb{E}[f_t(S^t)]$ over the time horizon $T$, where the expectation is taken from the randomness of the environment and the possible random algorithm of the player. In this paper, we are interested in online algorithms that are comparable to the solution of the offline greedy algorithm, namely the greedy sequence $\sigma^{\mathsf{G}} = \langle G_0, G_1, \ldots, G_{m^{\mathsf{G}}} \rangle$. Thus, the objective is to minimize the *greedy regret* defined as

$$R^{\mathsf{G}}(T) := T \cdot \mathbb{E}[f_1(G_{m^{\mathsf{G}}})] - \sum_{t=1}^{T} \mathbb{E}\left[f_t(S^t)\right]. \tag{1}$$

Given $\epsilon \geq 0$, we define the $\epsilon$-*quasi greedy regret* as

$$R^{\mathsf{Q}}(T) := T \cdot \mathbb{E}[f_1(Q_{m^{\mathsf{Q}}})] - \sum_{t=1}^{T} \mathbb{E}\left[f_t(S^t)\right], \tag{2}$$

where $\sigma^{\mathsf{Q}} = \langle Q_0, Q_1, \ldots, Q_{m^{\mathsf{Q}}} \rangle$ is the minimum $\epsilon$-quasi greedy sequence.

We remark that if the offline greedy algorithm provides an $\alpha$-approximation solution (with $0 < \alpha \leq 1$), then the greedy regret (or $\epsilon$-quasi greedy regret) also provides $\alpha$-approximation regret, which is the regret comparing to the $\alpha$ fraction of the optimal solution, as defined in [10].

In the rest of the paper, our goal is to design the player's policy that is comparable to the offline greedy, in other words, $R^{\mathsf{G}}(T)/T = \overline{f}(G_{m^{\mathsf{G}}}) - \frac{1}{T} \sum_{t=1}^{T} \mathbb{E}[f_t(S^t)] = o(1)$. Thus, to achieve sublinear greedy regret $R^{\mathsf{G}}(T) = o(T)$ is our main focus.

## 3 The Online Greedy and Algorithm OG-UCB

In this section, we propose our Online Greedy (OG) algorithm with the UCB policy to minimize the greedy regret (defined in (1)).

For any arm $a = e|S \in \mathcal{A}$, playing $a$ at each time $t$ yields the marginal reward as a random variable $X_t(a) = f_t(a)$, in which the random event $\omega_t \in \Omega$ is i.i.d., and we denote $\mu(a)$ as its true mean (i.e.,

---

**Algorithm 1** OG

---

**Require:** MaxOracle
1: **for** $t = 1, 2, \ldots$ **do**
2:      $S_0 \leftarrow \emptyset$;   $k \leftarrow 0$;   $h_0 \leftarrow$ true
3:      **repeat**                                            ▷ online greedy procedure
4:          $A \leftarrow \{e|S_k : \forall e \in \mathcal{N}(S_k)\}$;   $t' \leftarrow \sum_{a \in A} N(a) + 1$
5:          $(s_{k+1}|S_k, h_k) \leftarrow \mathrm{MaxOracle}\left(A, \hat{X}(\cdot), N(\cdot), t'\right)$         ▷ find the current maximal
6:          $S_{k+1} \leftarrow S_k \cup \{s_{k+1}\}$;   $k \leftarrow k+1$
7:      **until** $\mathcal{N}(S_k) = \emptyset$                             ▷ until a maximal sequence is found
8:      Play sequence $\sigma^t \leftarrow \langle S_0, \ldots, S_k \rangle$, observe $\{f_t(S_0), \ldots, f_t(S_k)\}$, and gain $f_t(S_k)$.
9:      **for all** $i = 1, 2, \ldots, k$ **do**         ▷ update according to signals from MaxOracle
10:          **if** $h_0, h_1, \cdots, h_{i-1}$ **are all** true **then**
11:              Update $\hat{X}(s_i|S_{i-1})$ and $N(s_i|S_{i-1})$ according to (3).

---

**Subroutine 2** UCB$(A, \hat{X}(\cdot), N(\cdot), t)$ to implement MaxOracle

---

**Setup:** confidence radius $\mathrm{rad}_t(a) := \sqrt{\frac{3 \ln t}{2N(a)}}$, for each $a \in A$
1: **if** $\exists a \in A$, $\hat{X}(a)$ is not initialized **then**              ▷ break ties arbitrarily
2:      **return** $(a, \mathsf{true})$                                   ▷ to initialize arms
3: **else**                                                     ▷ apply UCB's rule
4:      $I_t^+ \leftarrow \arg\max_{a \in A} \left\{ \hat{X}(a) + \mathrm{rad}_t(a) \right\}$, and **return** $(I_t^+, \mathsf{true})$

---

$\mu(a) := \mathbb{E}[X_1(a)]$). Let $\hat{X}(a)$ be the empirical mean for the marginal reward of $a$, and $N(a)$ be the counter of the plays. More specifically, denote $\hat{X}_t(a)$ and $N_t(a)$ for particular $\hat{X}(a)$ and $N(a)$ at the beginning of the time step $t$, and they are evaluated as follows:

$$\hat{X}_t(a) = \frac{\sum_{i=1}^{t-1} f_i(a) \mathbb{I}_i(a)}{\sum_{i=1}^{t-1} \mathbb{I}_i(a)}, \quad N_t(a) = \sum_{i=1}^{t-1} \mathbb{I}_i(a), \tag{3}$$

where $\mathbb{I}_i(a) \in \{0, 1\}$ indicates whether $a$ is updated at time $i$. In particular, assume that our algorithm is lazy-initialized so that each $\hat{X}(a)$ and $N(a)$ is 0 by default, until $a$ is played.

The *Online Greedy* algorithm (OG) proposed in Algorithm 1 serves as a meta-algorithm allowing different implementations of Subroutine MaxOracle. For every time $t$, OG calls MaxOracle (Line 5, to be specified later) to find the local maximal phase by phase, until the decision sequence $\sigma^t$ is made. Then, it plays sequence $\sigma^t$, observes feedbacks and gains the reward (Line 8). Meanwhile, OG collects the Boolean signals $(h_k)$ from MaxOracle during the greedy process (Line 5), and update estimators $\hat{X}(\cdot)$ and $N(\cdot)$ according to those signals (Line 10). On the other hand, MaxOracle takes accessible arms $A$, estimators $\hat{X}(\cdot), N(\cdot)$, and counted time $t'$, and returns an arm from $A$ and signal $h_k \in \{\mathsf{true}, \mathsf{false}\}$ to instruct OG whether to update estimators for the following phase.

The classical UCB [3] can be used to implement MaxOracle, which is described in Subroutine 2. We term our algorithm OG, in which MaxOracle is implemented by Subroutine 2 UCB, as Algorithm OG-UCB. A few remarks are in order: First, Algorithm OG-UCB chooses an arm with the highest upper confidence bound for each phase. Second, the signal $h_k$ is *always* true, meaning that OG-UCB always update empirical means of arms along the decision sequence. Third, because we use lazy-initialized $\hat{X}(\cdot)$ and $N(\cdot)$, the memory is allocated only when it is needed.

### 3.1 Regret Bound of OG-UCB

For any feasible set $S$, define the *greedy element for $S$* as $g_S^* := \arg\max_{e \in \mathcal{N}(S)} \overline{f}(e|S)$, and we use $\mathcal{N}_-(S) := \mathcal{N}(S) \setminus \{g_S^*\}$ for convenience. Denote $\mathcal{F}^\dagger := \{S \in \mathcal{F} : S \text{ is maximal}\}$ as the collection of all maximal feasible sets in $\mathcal{F}$. We use the following gaps to measure the performance of the algorithm.

**Definition 3.1** (Gaps). The *gap* between the maximal greedy feasible set $G_{m^G}$ and any $S \in \mathcal{F}$ is defined as $\Delta(S) := \overline{f}(G_{m^G}) - \overline{f}(S)$ if it is positive, and 0 otherwise. We define the *maximum gap* as $\Delta_{\max} = \overline{f}(G_{m^G}) - \min_{S \in \mathcal{F}^\dagger} \overline{f}(S)$, which is the worst penalty for any maximal feasible set. For any arms $a = e|S \in \mathcal{A}$, we define the *unit gap* of $a$ (i.e., the gap for one phase) as

$$\Delta(a) = \Delta(e|S) := \begin{cases} \overline{f}(g_S^*|S) - \overline{f}(e|S), & e \neq g_S^* \\ \overline{f}(g_S^*|S) - \max_{e' \in \mathcal{N}_-(S)} \overline{f}(e'|S), & e = g_S^* \end{cases}. \tag{4}$$

For any arms $a = e|S \in \mathcal{A}$, we define the *sunk-cost gap* (irreversible once selected) as

$$\Delta^*(a) = \Delta^*(e|S) := \max \left\{ \overline{f}(G_{m^G}) - \min_{V:V \in \mathcal{F}^\dagger, S \cup \{e\} \prec V} \overline{f}(V), 0 \right\}, \tag{5}$$

where for two feasible sets $A$ and $B$, $A \prec B$ means that $A$ is a prefix of $B$ in some decision sequence, that is, there exists a decision sequence $\sigma = \langle S_0 = \emptyset, S_1, \ldots, S_k \rangle$ such that $S_k = B$ and for some $j < k$, $S_j = A$. Thus, $\Delta^*(e|S)$ means the largest gap we may have after we have fixed our prefix selection to be $S \cup \{e\}$, and is upper bounded by $\Delta_{\max}$.

**Definition 3.2** (Decision frontier). For any decision sequence $\sigma = \langle S_0, S_1, \ldots, S_k \rangle$, define *decision frontier* $\Gamma(\sigma) := \bigcup_{i=1}^k \{e|S_{i-1} : e \in \mathcal{N}(S_{i-1})\} \subseteq \mathcal{A}$ as the arms need to be explored in the decision sequence $\sigma$, and $\Gamma_-(\sigma) := \bigcup_{i=1}^k \{e|S_{i-1} : \forall e \in \mathcal{N}_-(S_{i-1})\}$ similarly.

**Theorem 3.1** (Greedy regret bound). *For any time $T$, Algorithm* OG-UCB *(Algorithm 1 with Subroutine 2) can achieve the greedy regret*

$$R^G(T) \leq \sum_{a \in \Gamma_-(\sigma^G)} \left( \frac{6\Delta^*(a) \cdot \ln T}{\Delta(a)^2} + \left( \frac{\pi^2}{3} + 1 \right) \Delta^*(a) \right), \tag{6}$$

*where $\sigma^G$ is the greedy decision sequence.*

When $m = 1$, the above theorem immediately recovers the regret bound of the classical UCB [3] (with $\Delta^*(a) = \Delta(a)$). The greedy regret is bounded by $O\left(\frac{mW\Delta_{\max}\log T}{\Delta^2}\right)$ where $\Delta$ is the minimum unit gap ($\Delta = \min_{a \in \mathcal{A}} \Delta(a)$), and the memory cost is at most proportional to the regret. For a special class of linear bandits, a simple extension where we treat arms $e|S$ and $e|S'$ as the same can make OG-UCB essentially the same as OMM in [20], while the regret is $O(\frac{n}{\Delta} \log T)$ and the memory cost is $O(n)$ (cf. Appendix F.1 of the supplementary material).

# 4 Relaxing the Greedy Sequence with $\epsilon$-Error Tolerance

In this section, we propose an online algorithm called OG-LUCB, which learns an $\epsilon$-quasi greedy sequence, with the goal of minimizing the $\epsilon$-quasi greedy regret (in (2)). We learn $\epsilon$-quasi-greedy sequences by a *first-explore-then-exploit policy*, which utilizes results from PAC learning with a fixed confidence setting. In Section 4.1, we implement MaxOracle via the LUCB policy, and derive its exploration time; we then assume the knowledge of time horizon $T$ in Section 4.2, and analyze the $\epsilon$-quasi greedy regret; and in Section 4.3, we show that the assumption of knowing $T$ can be further removed.

## 4.1 OG with a first-explore-then-exploit policy

Given $\epsilon \geq 0$ and failure probability $\delta \in (0, 1)$, we use Subroutine 3 LUCB$_{\epsilon,\delta}$ to implement the subroutine MaxOracle in Algorithm OG. We call the resulting algorithm OG-LUCB$_{\epsilon,\delta}$. Specifically, Subroutine 3 is adapted from CLUCB-PAC in [9], and specialized to explore the top-one element in the support of $[0, 1]$ (i.e., set $R = \frac{1}{2}$, $\text{width}(\mathcal{M}) = 2$ and $\text{Oracle} = \arg\max$ in [9]). Assume that $I^{\text{exploit}}(\cdot)$ is lazy-initialized. For each greedy phase, the algorithm first explores each arm in $A$ in the *exploration* stage, during which the return flag (the second return field) is always false; when the optimal one is found (initialize $I^{\text{exploit}}(A)$ with $\hat{I}_t$), it sticks to $I^{\text{exploit}}(A)$ in the *exploitation* stage for the subsequent time steps, and return flag for this phase becomes true. The main algorithm OG then uses these flags in such a way that it updates arm estimates for phase $i$ if any only if all phases

---

**Subroutine 3** $\text{LUCB}_{\epsilon,\delta}(A, \hat{X}(\cdot), N(\cdot), t)$ to implement MaxOracle

---

**Setup:** $\text{rad}_t(a) := \sqrt{\frac{\ln(4Wt^3/\delta)}{2N(a)}}$, for each $a \in A$; $I^{\text{exploit}}(\cdot)$ to cache arms for exploitation;

1: **if** $I^{\text{exploit}}(A)$ is initialized **then return** $(I^{\text{exploit}}(A), \text{true})$          $\triangleright$ in the *exploitation* stage
2: **if** $\exists a \in A, \hat{X}(a)$ is not initialized **then**          $\triangleright$ break ties arbitrarily
3:      **return** $(a, \text{false})$          $\triangleright$ to initialize arms
4: **else**
5:      $\hat{I}_t \leftarrow \arg\max_{a \in A} \hat{X}(a)$
6:      $\forall a \in A, X'(a) \leftarrow \begin{cases} \hat{X}(a) + \text{rad}_t(a), & a \neq \hat{I}_t \\ \hat{X}(a) - \text{rad}_t(a), & a = \hat{I}_t \end{cases}$          $\triangleright$ perturb arms
7:      $I'_t \leftarrow \arg\max_{a \in A} X'(a)$
8:      **if** $X'(I'_t) - X'(\hat{I}_t) > \epsilon$ **then**          $\triangleright$ not separated
9:          $I''_t \leftarrow \arg\max_{i \in \{\hat{I}_t, I'_t\}} \text{rad}_t(i)$, and **return** $(I''_t, \text{false})$    $\triangleright$ in the *exploration* stage
10:      **else**          $\triangleright$ separated
11:          $I^{\text{exploit}}(A) \leftarrow \hat{I}_t$          $\triangleright$ initialize $I^{\text{exploit}}(A)$ with $\hat{I}_t$
12:          **return** $(I^{\text{exploit}}(A), \text{true})$          $\triangleright$ in the *exploitation* stage

---

for $j < i$ are already in the exploitation stage. This avoids maintaining useless arm estimates and is a major memory saving comparing to OG-UCB.

In Algorithm OG-LUCB$_{\epsilon,\delta}$, we define the *total exploration time* $T^{\mathsf{E}} = T^{\mathsf{E}}(\delta)$, such that for any time $t \geq T^{\mathsf{E}}$, OG-LUCB$_{\epsilon,\delta}$ is in the exploitation stage for all greedy phases encountered in the algorithm. This also means that after time $T^{\mathsf{E}}$, in every step we play the same maximal decision sequence $\sigma = \langle S_0, S_1, \cdots, S_k \rangle \in \mathcal{F}^{k+1}$, which we call a *stable sequence*. Following a common practice, we define the *hardness coefficient* with prefix $S \in \mathcal{F}$ as

$$H_S^\epsilon := \sum_{e \in \mathcal{N}(S)} \frac{1}{\max\{\Delta(e|S)^2, \epsilon^2\}}, \text{ where } \Delta(e|S) \text{ is defined in (4).} \tag{7}$$

**Rewrite definitions with respect to the $\epsilon$-quasi regret.** Recall that $\sigma^{\mathsf{Q}} = \langle Q_0, Q_1, \ldots, Q_{m^{\mathsf{Q}}} \rangle$ is the minimum $\epsilon$-quasi greedy sequence. In this section, we rewrite the gap $\Delta(S) := \max\{\overline{f}(Q_{m^{\mathsf{Q}}}) - \overline{f}(S), 0\}$ for any $S \in \mathcal{F}$, the maximum gap $\Delta_{\max} := \overline{f}(Q_{m^{\mathsf{Q}}}) - \min_{S \in \mathcal{F}^\dagger} \overline{f}(S)$, and $\Delta^*(a) = \Delta^*(e|S) := \max\{\overline{f}(Q_{m^{\mathsf{Q}}}) - \min_{V:V \in \mathcal{F}^\dagger, S \cup \{e\} \prec V} \overline{f}(V), 0\}$, for any arm $a = e|S \in \mathcal{A}$.

The following theorem shows that, with high probability, we can find a stable $\epsilon$-quasi greedy sequence, and the total exploration time is bounded.

**Theorem 4.1** (High probability exploration time). *Given any $\epsilon \geq 0$ and $\delta \in (0,1)$, suppose after the total exploration time $T^{\mathsf{E}} = T^{\mathsf{E}}(\delta)$, Algorithm OG-LUCB$_{\epsilon,\delta}$ (Algorithm 1 with Subroutine 3) sticks to a stable sequence $\sigma = \langle S_0, S_1, \cdots, S_{m'} \rangle$ where $m'$ is its length. With probability at least $1 - m\delta$, the following claims hold: (1) $\sigma$ is an $\epsilon$-quasi greedy sequence; (2) The total exploration time satisfies that $T^{\mathsf{E}} \leq 127 \sum_{k=0}^{m'-1} H_S^\epsilon \ln(1996 W H_S^\epsilon / \delta)$,*

## 4.2 Time Horizon $T$ is Known

Knowing time horizon $T$, we may let $\delta = \frac{1}{T}$ in OG-LUCB$_{\epsilon,\delta}$ to derive the $\epsilon$-quasi regret as follows.

**Theorem 4.2.** *Given any $\epsilon \geq 0$. When total time $T$ is known, let Algorithm OG-LUCB$_{\epsilon,\delta}$ run with $\delta = \frac{1}{T}$. Suppose $\sigma = \langle S_0, S_1, \cdots, S_{m'} \rangle$ is the sequence selected at time $T$. Define function $R^{\mathsf{Q},\sigma}(T) := \sum_{e|S \in \Gamma(\sigma)} \Delta^*(e|S) \min\left\{\frac{127}{\Delta(e|S)^2}, \frac{113}{\epsilon^2}\right\} \ln(1996 W H_S^\epsilon T) + \Delta_{\max} m$, where $m$ is the largest length of a feasible set and $H_S^\epsilon$ is defined in (7). Then, the $\epsilon$-quasi regret satisfies that $R^{\mathsf{Q}}(T) \leq R^{\mathsf{Q},\sigma}(T) = O(\frac{Wm\Delta_{\max}}{\max\{\Delta^2, \epsilon^2\}} \log T)$, where $\Delta$ is the minimum unit gap.*

In general, the two bounds (Theorem 3.1 and Theorem 4.2) are for different regret metrics, thus can not be directly compared. When $\epsilon = 0$, OG-UCB is slightly better only in the constant before $\log T$. On other hand, when we are satisfied with $\epsilon$-quasi greedy regret, OG-LUCB$_{\epsilon,\delta}$ may work better for

---

**Algorithm 4** OG-LUCB-R (i.e., OG-LUCB with Restart)

---

**Require:** $\epsilon$
 1: **for** epoch $\ell = 1, 2, \cdots$ **do**
 2:     Clean $\hat{X}(\cdot)$ and $N(\cdot)$ for all arms, and restart OG-LUCB$_{\epsilon,\delta}$ with $\delta = \frac{1}{\phi_\ell}$ (defined in (8)).
 3:     Run OG-LUCB$_{\epsilon,\delta}$ for $\phi_\ell$ time steps. (exit halfway, if the time is over.)

---

some large $\epsilon$, for the bound takes the maximum (in the denominator) of the problem-dependent term $\Delta(e|S)$ and the fixed constant $\epsilon$ term, and the memory cost is only $O(mW)$.

### 4.3 Time Horizon $T$ is not Known

When time horizon $T$ is not known, we can apply the "squaring trick", and restart the algorithm for each epoch as follows. Define *the duration of epoch $\ell$ as $\phi_\ell$*, and its *accumulated time as $\tau_\ell$*, where

$$\phi_\ell := e^{2^\ell}; \quad \tau_\ell := \begin{cases} 0, & \ell = 0 \\ \sum_{s=1}^{\ell} \phi_s, & \ell \geq 1 \end{cases}. \tag{8}$$

For any time horizon $T$, define the *final epoch $K = K(T)$* as the epoch where $T$ lies in, that is $\tau_{K-1} < T \leq \tau_K$. Then, our algorithm OG-LUCB-R is proposed in Algorithm 4. The following theorem shows that the $O(\log T)$ $\epsilon$-quasi regret still holds, with a slight blowup of the constant hidden in the big O notation (For completeness, the explicit constant before $\log T$ can be found in Theorem D.7 of the supplementary material).

**Theorem 4.3.** *Given any $\epsilon \geq 0$. Use $\phi_\ell$ and $\tau_\ell$ defined in (8), and function $R^{\mathsf{Q},\sigma}(T)$ defined in Theorem 4.2. In Algorithm OG-LUCB-R, suppose $\sigma^{(\ell)} = \langle S_0^{(\ell)}, S_1^{(\ell)}, \cdots, S_{m^{(\ell)}}^{(\ell)} \rangle$ is the sequence selected by the end of $\ell$-th epoch of OG-LUCB$_{\epsilon,\delta}$, where $m^{(\ell)}$ is its length. For any time $T$, denote final epoch as $K = K(T)$ such that $\tau_{K-1} < T \leq \tau_K$, and the $\epsilon$-quasi regret satisfies that $R^{\mathsf{Q}}(T) \leq \sum_{\ell=1}^{K} R^{\mathsf{Q},\sigma^{(\ell)}}(\phi_\ell) = O\left(\frac{Wm\Delta_{\max}}{\max\{\Delta^2, \epsilon^2\}} \log T\right)$, where $\Delta$ is the minimum unit gap.*

## 5 Lower Bound on the Greedy Regret

Consider a problem of selecting one element each from $m$ bandit instances, and the player sequentially collects prize at every phase. For simplicity, we call it the *prize-collecting problem*, which is defined as follows: For each bandit instance $i = 1, 2, \ldots, m$, denote set $E_i = \{e_{i,1}, e_{i,2}, \ldots, e_{i,W}\}$ of size $W$. The accessible set system is defined as $(E, \mathcal{F})$, where $E = \bigcup_{i=1}^{m} E_i$, $\mathcal{F} = \cup_{i=1}^{m} \mathcal{F}_i \cup \{\emptyset\}$, and $\mathcal{F}_i = \{S \subseteq E : |S| = i, \forall k : 1 \leq k \leq i, |S \cap E_k| = 1\}$. The reward function $f : \mathcal{F} \times \Omega \to [0, m]$ is non-decreasing in the first parameter, and the form of $f$ is unknown to the player. Let minimum unit gap $\Delta := \min\left\{\overline{f}(g_S^*|S) - \overline{f}(e|S) : \forall S \in \mathcal{F}, \forall e \in \mathcal{N}_-(S)\right\} > 0$, where its value is also unknown to the player. The objective of the player is to minimize the greedy regret.

Denote the greedy sequence as $\sigma^{\mathsf{G}} = \langle G_0, G_1, \cdots, G_m \rangle$, and the greedy arms as $\mathcal{A}^{\mathsf{G}} = \{g_{G_{i-1}}^*|G_{i-1} : \forall i = 1, 2, \cdots, W\}$. We say an algorithm is *consistent*, if the sum of playing all arms $a \in \mathcal{A} \setminus \mathcal{A}^{\mathsf{G}}$ is in $o(T^\eta)$, for any $\eta > 0$, i.e., $\mathbb{E}[\sum_{a \in \mathcal{A} \setminus \mathcal{A}^{\mathsf{G}}} N_T(a)] = o(T^\eta)$.

**Theorem 5.1.** *For any consistent algorithm, there exists a problem instance of the prize-collecting problem, as time $T$ tends to $\infty$, for any minimum unit gap $\Delta \in (0, \frac{1}{4})$, such that $\Delta^2 \geq \frac{2}{3W^{\xi m-1}}$ for some constant $\xi \in (0, 1)$, the greedy regret satisfies that $R^{\mathsf{G}}(T) = \Omega\left(\frac{mW \ln T}{\Delta^2}\right)$.*

We remark that the detailed problem instance and the greedy regret can be found in Theorem E.2 of the supplementary material. Furthermore, we may also restrict the maximum gap $\Delta_{\max}$ to $\Theta(1)$, and the lower bound $R^{\mathsf{G}}(T) = \Omega(\frac{mW\Delta_{\max} \ln T}{\Delta^2})$, for any sufficiently large $T$. For the upper bound, OG-UCB (Theorem 3.1) gives that $R^{\mathsf{G}}(T) = O(\frac{mW\Delta_{\max}}{\Delta^2} \log T)$, Thus, our upper bound of OG-UCB matches the lower bound within a constant factor.

**Acknowledgments** Jian Li was supported in part by the National Basic Research Program of China grants 2015CB358700, 2011CBA00300, 2011CBA00301, and the National NSFC grants 61202009, 61033001, 61361136003.

## Footnotes

[1]Therefore, the optimal solution is a maximal decision sequence.

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
