[Supplementary Material · nips2015_full.pdf]

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

| Symbols in the main text | Definition |
|---|---|
| $(E, \mathcal{F})$ | the ground set, and the collection of all feasible sets in Section 2 |
| $\mathcal{N}(S)$ | the accessible set from prefix $S$ in Section 2 |
| $n$ | the size of ground set $|E|$ in Section 2 |
| $m$ | the maximal length of any feasible set in Section 2 |
| $W$ | the maximal width of any accessible set in Section 2 |
| $a = e|S$, $\mathcal{A}$ | one arm, and the arm space in Section 2 |
| $\sigma = \langle S_0, \cdots, S_k \rangle$ | a decision sequence in Section 2 |
| $f(\cdot, \cdot)$, $f_t(\cdot)$ | the reward function, and its shorthand in Section 2 |
| $\bar{f}(\cdot)$ | the expected reward function in Section 2 |
| $\sigma^{\mathsf{G}} = \langle G_0, \cdots, G_{m^{\mathsf{G}}} \rangle$ | the greedy sequence where $m^{\mathsf{G}}$ is its length in Section 2.1 |
| $\sigma^{\mathsf{Q}} = \langle Q_0, \cdots, Q_{m^{\mathsf{Q}}} \rangle$ | the minimum $\epsilon$-quasi greedy sequence where $m^{\mathsf{Q}}$ is its length in Section 2.1 |
| $\Gamma(\sigma)$ | the decision frontier of $\sigma$ in Section 3.1 |
| $\Gamma_-(\sigma)$ | the decision frontier of $\sigma$, excluding all greedy elements in Section 3.1 |
| $R(T)$ | the cumulative regret in Section 2.2 |
| $R^{\mathsf{G}}(T)$, $R^{\mathsf{Q}}(T)$, | the greedy regret ((1)), the $\epsilon$-quasi greedy regret ((2)) |
| $R^{\alpha}(T)$ | the $\alpha$-approximation regret discussed in Section 2.2 (formally defined in Definition B.1) |
| $\mathcal{F}^{\dagger}$ | the collection of all maximal feasible sets in Section 3.1 |
| $\hat{X}(a)$, $N(a)$, $X(a)$ | the mean estimator of $\{f_t(a)\}_{t=1}^{\infty}$, the counter, and the true mean in Section 3 |
| $\hat{X}_t(a)$, $N_t(a)$ | the particular $\hat{X}(a)$ and $N(a)$ at the beginning of the time step $t$, in Section 3 |
| $g_S^*$ | the greedy element of prefix $S$ in Section 3.1 |
| $\mathcal{N}_-(S)$ | the accessible set from prefix $S$, excluding the greedy element, in Section 3.1 |
| $\Delta(S)$ | the gap between $\sigma^{\mathsf{G}}$ (or $\sigma^{\mathsf{Q}}$) and $S$ in Definition 3.1 (or rewritten in Section 4.1) |
| $\Delta_{\max}$ | the maximum gap of $\Delta(S)$ defined in Definition 3.1 (or rewritten in Section 4.1) |
| $\Delta(a)$ (or $\Delta(e|S)$) | the unit gap of arm $a = e|S$ defined in (4) |
| $\Delta^*(a)$ (or $\Delta^*(e|S)$) | the maximum gap of selecting $a = e|S$ defined in (5) (rewritten in Section 4.1) |
| $H_S^{\epsilon}$ | the hardness coefficient in Section 4.1 |
| $T^{\mathsf{E}}$ | the total exploration time for OG-LUCB$_{\epsilon,\delta}$ until a stable sequence is found |
| $N_t^{\mathsf{E}}(a)$ | the counter of playing arm $a$ during the exploration before time $t$ in Section 4.1 |
| $\phi_\ell$, $\tau_\ell$ | the duration of $\ell$-th epoch, and its accumulated time defined in (8) |
| $K = K(T)$ | the final epoch of time horizon $T$ defined in Section 4.3 |
| $R^{\mathsf{Q},\sigma}(T)$ | an intermediate form of the $\epsilon$-quasi regret for sequence $\sigma$ defined in Theorem 4.2 |

Table 1: List of symbols in the main text.

## Appendix

The appendix is organized as follows.

Above all, for ease of reading, we list the symbols and their definitions in the main text in Table 1.

In Appendix A, we show that any feasible set in $\mathcal{F}$ is accessible by a decision sequence, and demonstrate the concrete decision classes in our model with a few examples.

In Appendix B, we formally define $\alpha$-*approximation regret*, and give some simple propositions to show that such a regret can be derived immediately from the greedy regret and the $\epsilon$-quasi greedy regret if the offline solution is an $\alpha$-approximation solution.

In Appendix C, we first establish lemmas by the set decomposition for Algorithm OG-UCB, and prove that OG-UCB can achieve the problem-dependent $O(\log T)$ greedy regret.

In Appendix D, we relax the greedy sequence to tolerate $\epsilon$-error. We prove the high-probability exploration time of for Algorithm OG-LUCB$_{\epsilon,\delta}$ in the beginning. By letting $\delta = \frac{1}{T}$ with the known time horizon $T$, we show that the first-explore-then-exploit policy of Algorithm OG-LUCB$_{\epsilon,\delta}$ (associate $\delta = 1/T$) achieves the problem-dependent $O(\log T)$ $\epsilon$-quasi regret bound. Then, we show that Algorithm OG-LUCB-R utilizing OG-LUCB$_{\epsilon,\delta}$ can remove the dependence on $T$, and the $O(\log T)$ bound holds with a slight compensation for its constant.

In Appendix E, we construct a problem instance of the prize-collecting problem, and show the upper bound is tight with the lower bound (up to a constant factor).

In Appendix F, we show a simple extension to recover $O(\frac{n}{\Delta} \log T)$ bound for a linear bandit with matroid constraints, and discuss an extension of our model to the Knapsack problem.

In Appendix G, we first apply our algorithms to the top-$m$ selection problem with a consistent function and the online submodular problem, and show the $\alpha$-approximation regret when the offline greedy yields an $\alpha$-approximation solution. Then we discuss the application to the online version of the influence maximization problem and the probabilistic set cover problem, which may remove the assumption on one particular diffusion model and avoid the issue of model misspecification.

In Appendix H, we evaluate the lower bound for the prize-collecting problem, and compare the regret of OG-UCB and the lower bound numerically.

## A  The Accessible Set System $(E, \mathcal{F})$

An accessible set system $(E, \mathcal{F})$ (in Section 2) satisfies two axioms:

*(A1: Triviality axiom).* $\emptyset \in \mathcal{F}$;

*(A2: Accessibility axiom).* If $S \in \mathcal{F}$ and $S \neq \emptyset$, then there exists some $e$ in $E$, s.t., $S \setminus \{e\} \in \mathcal{F}$.

We claim that any set $S \in \mathcal{F}$ can be obtained by adding one element at a time from the empty set. Formally, we have the following fact.

**Lemma A.1** (Accessibility of any feasible set). *For any set $S \in \mathcal{F}$, denote $k = |S|$. There exists a sequence $\langle S_0, S_1, \cdots, S_k \rangle$, such that: (1) $S_0, S_1, \cdots, S_k \in \mathcal{F}$; (2) $\emptyset = S_0 \subset S_1 \subset \cdots \subset S_k = S$; (3) $|S_i| - |S_{i-1}| = 1$, for each $i = 1, 2, \ldots, k$.*

*Proof.* If $S = \emptyset$, it is obvious that the sequence $\{S_0\}$ satisfies the above three properties, therefore we only focus on a non-empty $S$, that is $k \geq 1$.

For any set $S \in \mathcal{F}$, let $S_k = S$. From Axiom A2, we know that there exists some $e \in E$, $S_k \setminus \{e\} \in \mathcal{F}$, thus we can denote $S_{k-1} = S_k \setminus \{e\}$. We can carry on this procedure, and apply Axiom A2 iteratively. Then we can get $S_i$, for each $i = k - 1, k - 2, \ldots, 0$. Therefore, the sequence $\langle S_0, S_1, \cdots, S_k \rangle$ we found satisfies the three properties, which ends the proof. $\square$

Our characterization of accessible set system $(E, \mathcal{F})$ encompasses greedoids and matroids as special cases, which is studied in the previous literature [5, 17]. More specifically, from [17], we know that:

**Definition A.1** (Greedoid). A greedoid is an accessible set system $(E, \mathcal{F})$ satisfying:

*(A3: Augmentation axiom).* For all $S, T \in \mathcal{F}$ such that $|S| > |T|$, there is an $x \in S \setminus T$ such that $T \cup \{x\} \in \mathcal{F}$.

**Definition A.2** (Matroid). A matroid is a greedoid $(E, \mathcal{F})$ satisfying:

*(A4: Hereditary axiom).* If $S \in \mathcal{F}$ and $T \subseteq S$, then $T \in \mathcal{F}$.

In addition, we list a few concrete examples that can fit into our model.

**Example A.1** (Top-$m$ selection). In accessible set system $(E, \mathcal{F})$, $\mathcal{F}$ contains all sets $S \subseteq E$, where $|S| \leq m$. Therefore, a maximal feasible set is a subset of size $m$.

**Example A.2** (Spanning Tree). Given a graph $\mathcal{G} = (V, E)$, a forest $F$ in $\mathcal{G}$ is a subset of edges in $E$ that does not contains a cycle. The corresponding accessible set system is $(E, \mathcal{F})$, where $\mathcal{F} = \{F \subseteq E : F \text{ is a forest}\}$. The maximal set of a spanning tree constraint is a forest that cannot include more edges.

**Example A.3** (Gaussian Elimination Greedoids [32]). Let $M = (m_{ij}) \in \mathbb{K}^{m \times n}$ be a $n \times m$ matrix over an arbitrary field $\mathbb{K}$. The accessible set system is $(E, \mathcal{F})$ where $E = \{1, 2, \cdots, n\}$ and $\mathcal{F} = \{A \subseteq E : \text{the submatrix } M_{\{1,2,\cdots,|A|\}, A} \text{ is non-singular}\}$. It corresponds to the procedure of performing Gaussian Elimination on rows, which give rise to the sequence of column indices. A maximal feasible set $A$ makes a submatrix that with the same rank as matrix $M$.

**Example A.4** (Knapsack)**.** Let the ground set be $E = \{e_1, e_2, \cdots, e_n\}$ and the cost function be $\lambda : E \to \mathbb{R}_{>0}$. Given the budget $B$ for the knapsack problem, the accessible set system is $(E, \mathcal{F})$, where $\mathcal{F} = \{S \subseteq E : \sum_{e \in S} \lambda(e) \leq B\}$. $(E, \mathcal{F})$ corresponding to the knapsack problem is neither a greedoid nor a matroid, but nevertheless still an accessible set system. A maximal feasible set $S$ is the one that cannot include any more element in $E \setminus S$ without violating the budget constraint, and maximal feasible sets may have different lengths.

## B   Some Simple Propositions

We first formally define $\alpha$-*approximation regret*, the regret comparing to the $\alpha$ fraction of the optimal solution (as defined in [11]), in the following.

**Definition B.1** ($\alpha$-approximation regret)**.** Let $S^* = \arg\max_{S \in \mathcal{F}} \mathbb{E}[f_1(S)]$, and denote $\sigma^t = \langle S_0^t, S_1^t, \ldots, S_{m^t}^t \rangle$, where $S^t := S_{m^t}^t$, as the decision sequence selected for any time $t = 1, 2, \cdots, T$. We define the $\alpha$-approximation regret ($0 < \alpha \leq 1$) as

$$R^\alpha(T) := \alpha T \cdot \mathbb{E}[f_1(S^*)] - \sum_{t=1}^T \mathbb{E}[f_t(S^t)]. \tag{9}$$

Then, the next two simple propositions show that the $\alpha$-approximation regret can be derived from the greedy regret and $\epsilon$-quasi greedy regret if the offline greedy algorithm achieves $\alpha$-approximation solution.

**Proposition B.1.** *For the greedy sequence $\sigma^{\mathsf{G}}$, if the maximal feasible set in $\sigma^{\mathsf{G}}$ is an $\alpha$-approximation for the offline problem, where $\alpha \in (0, 1]$, then $R^\alpha(T) \leq R^{\mathsf{G}}(T)$.*

*Proof.* For the greedy sequence $\sigma^{\mathsf{G}} := \langle G_0, G_1, \ldots, G_{m^{\mathsf{G}}} \rangle$. Since $G_{m^{\mathsf{G}}}$ is an $\alpha$-approximation solution, thus $\mathbb{E}[f_1(S^{\mathsf{G}})] = \overline{f}(S^{\mathsf{G}}) \geq \alpha \overline{f}(S^*) = \mathbb{E}[f_1(S^*)]$. Therefore, it follows that

$$R^{\mathsf{G}}(T) = T \cdot \mathbb{E}[f_1(S^{\mathsf{G}})] - \sum_{t=1}^T \mathbb{E}[f_t(S^t)] \geq T \cdot \alpha \mathbb{E}[f_1(S^*)] - \sum_{t=1}^T \mathbb{E}[f_t(S^t)] = R^\alpha(T).$$

$\square$

A similar proposition holds for $\epsilon$-quasi greedy sequences. Among all $\epsilon$-quasi greedy sequences, we use $\sigma^{\mathsf{Q}} := \langle Q_0, Q_1, \ldots, Q_{m^{\mathsf{Q}}} \rangle$ to denote the one with the minimum reward, where $m^{\mathsf{Q}}$ is its size.

**Proposition B.2.** *For an $\epsilon$-quasi greedy sequence $\sigma^{\mathsf{Q}} := \langle Q_0, Q_1, \ldots, Q_{m^{\mathsf{Q}}} \rangle$, if $Q_{m^{\mathsf{Q}}}$ is an $\alpha$-approximation for the offline problem, where $\alpha \in (0, 1]$, then $R^\alpha(T) \leq R^{\mathsf{Q}}(T)$.*

*Proof.* Since $Q_{m^{\mathsf{Q}}}$ is an $\alpha$-approximation, $\mathbb{E}[f_1(Q_{m^{\mathsf{Q}}})] = \overline{f}(Q_{m^{\mathsf{Q}}}) \geq \alpha \overline{f}(S^*) = \mathbb{E}[f_1(S^*)]$. Hence, we can get that

$$R^{\mathsf{Q}}(T) = T \cdot \mathbb{E}[f_1(Q_{m^{\mathsf{Q}}})] - \sum_{t=1}^T \mathbb{E}[f_t(S^t)] \geq T \cdot \alpha \mathbb{E}[f_1(S^*)] - \sum_{t=1}^T \mathbb{E}[f_t(S^t)] = R^\alpha(T).$$

$\square$

## C   Analysis for **OG-UCB** in Section 3

In this section, we analyze the greedy regret bound for Algorithm OG-UCB (Algorithm 1 + Subroutine 2). In Appendix C.1, we first derive a general form of the greedy regret bound. Then, in Appendix C.2, we analyze the expected number of playing each arm in the decision frontier, and then obtain the desired greedy regret bound for Algorithm OG-UCB.

## C.1 Set Decomposition

We first define some useful notations. For any time $t$, suppose Algorithm OG-UCB plays a maximal decision sequence $\sigma^t := \langle S_0^t, S_1^t, \ldots, S_{m^t}^t \rangle$ where $S^t := S_{m^t}^t$, and $S_i^t = S_{i-1}^t \cup \{s_i^t\}$ for each $i$. Denote the greedy sequence as $\sigma^{\mathsf{G}} := \langle G_0, G_1, \ldots, G_{m^{\mathsf{G}}} \rangle$, where $G_i = G_{i-1} \cup \{g_i\}$ for each $i$. For any time $t$ and any $i = 1, 2, \cdots$, define random event $\mathcal{O}_i^t := \{G_i = S_i^t\}$ (we define $\mathcal{O}_0^t = \{\emptyset = \emptyset\}$ always holds.) which means that OG-UCB finds the same prefix $S_i^t$ as the greedy sequence does. For ease of notation, let $\mathcal{O}_{0:k}^t := \bigcap_{i=0}^{k} \mathcal{O}_i^t$ for each $k \geq 0$.

Then, we can get the following lemma by De Morgan's laws:

**Lemma C.1** (Set-decomposition). *Fix any $k \geq 1$, then* $\overline{\bigcap_{i=1}^{k} \mathcal{O}_i^t} = \left( \overline{\mathcal{O}_1^t} \right) \cup \left( \mathcal{O}_1^t \cap \overline{\mathcal{O}_2^t} \right) \cup \cdots \cup \left( \left( \bigcap_{i=1}^{k-1} \mathcal{O}_i^t \right) \cap \overline{\mathcal{O}_k^t} \right) = \bigcup_{j=1}^{k} \left( \mathcal{O}_{0:j-1}^t \cap \overline{\mathcal{O}_j^t} \right)$, *and each term of its right-hand side is mutually exclusive.*

Notice that $\mathcal{O}_{1:j-1}^t \cap \overline{\mathcal{O}_j^t}$ means that the first $j-1$ prefixes coincide with the greedy sequence. i.e., $G_i = S_i^t$, for $i = 1, \cdots, j-1$; however, at $j$-th phase, it picks $s_j^t \neq g_j$, that is $s_j^t \in \mathcal{N}_-(G_{j-1})$. Now, we can write the greedy regret as follows.

**Lemma C.2.** *For any time $T$, the greedy regret of Algorithm OG-UCB satisfies that:*

$$R^{\mathsf{G}}(T) \leq \sum_{k=1}^{m^{\mathsf{G}}} \mathbb{E} \left[ \sum_{t=1}^{T} \Delta(S^t) \cdot \mathbb{I}\left\{ \mathcal{O}_{1:k-1}^t \cap \overline{\mathcal{O}_k^t} \right\} \right].$$

*Proof.* By definition, $\overline{f}(S) = \mathbb{E}_{\omega_1}[f_1(S)]$ and $\Delta(S) = \overline{f}(G_{m^{\mathsf{G}}}) - \overline{f}(S)$ for any $S \in \mathcal{F}$. Therefore, we have that

$$R^{\mathsf{G}}(T) = T \cdot \mathbb{E}\left[f_1(G_{m^{\mathsf{G}}})\right] - \sum_{t=1}^{T} \mathbb{E}\left[f_t(S^t)\right]$$

$$= T \cdot \overline{f}(G_{m^{\mathsf{G}}}) - \sum_{t=1}^{T} \mathbb{E}\left[\overline{f}(S^t)\right]$$

$$= \mathbb{E}\left[ \sum_{t=1}^{T} \Delta(S^t) \cdot \mathbb{I}\{G_{m^{\mathsf{G}}} \neq S^t\} \right].$$

From the definition, we know that if $\bigcap_{i=1}^{m^{\mathsf{G}}} \mathcal{O}_i^t$ occurs, then event $\{G_{m^{\mathsf{G}}} = S^t\}$ is true. As its contrapositive, $\{G_{m^{\mathsf{G}}} \neq S^t\}$ implies that $\overline{\cap_{i=1}^{m^{\mathsf{G}}} \mathcal{O}_i^t}$ occurs. Therefore, we can get

$$\mathbb{E}\left[ \sum_{t=1}^{T} \Delta(S^t) \cdot \mathbb{I}\{G_{m^{\mathsf{G}}} \neq S^t\} \right] \leq \mathbb{E}\left[ \sum_{t=1}^{T} \Delta(S^t) \cdot \mathbb{I}\left\{ \overline{\cap_{i=1}^{m^{\mathsf{G}}} \mathcal{O}_i^t} \right\} \right]$$

$$\leq \mathbb{E}\left[ \sum_{t=1}^{T} \Delta(S^t) \cdot \mathbb{I}\left\{ \bigcup_{k=1}^{m^{\mathsf{G}}} \left( \mathcal{O}_{0:k-1}^t \cap \overline{\mathcal{O}_k^t} \right) \right\} \right] \quad (10)$$

$$\leq \mathbb{E}\left[ \sum_{t=1}^{T} \Delta(S^t) \cdot \sum_{k=1}^{m^{\mathsf{G}}} \mathbb{I}\left\{ \mathcal{O}_{0:k-1}^t \cap \overline{\mathcal{O}_k^t} \right\} \right] \quad (11)$$

$$= \sum_{k=1}^{m^{\mathsf{G}}} \mathbb{E}\left[ \sum_{t=1}^{T} \Delta(S^t) \cdot \mathbb{I}\left\{ \mathcal{O}_{0:k-1}^t \cap \overline{\mathcal{O}_k^t} \right\} \right],$$

where (10) is due to Lemma C.1, and (11) is by the union bound. $\qquad\square$

## C.2 Proof of Theorem 3.1

We use the following well-known probability inequality.

**Fact C.3** (Chernoff-Hoeffding Bound). $Z_1, Z_2, \ldots, Z_m$ *are i.i.d. random variables supported on* $[0,1]$ *with mean* $\mu$. *Define the mean estimator as* $\hat{Z} = \frac{1}{m}\sum_{i=1}^{m} Z_m$. *For any* $a > 0$, *it holds that*

$$\mathbb{P}\left[\hat{Z} > \mu + \epsilon\right] \leq \exp\left\{-2\epsilon^2 m\right\}, \quad \mathbb{P}\left[\hat{Z} < \mu - \epsilon\right] \leq \exp\left\{-2\epsilon^2 m\right\}.$$

Recall that the unit gap for any arm $a = e|S$ (defined in (4)) is

$$\Delta(a) = \Delta(e|S) = \begin{cases} \overline{f}(g_S^*|S) - \overline{f}(e|S), & e \neq g_S^* \\ \overline{f}(e|S) - \max_{e' \in \mathcal{N}_-(S)} \overline{f}(e'|S), & e = g_S^* \end{cases}.$$

Define the threshold function for any arm $a = e|S \in \mathcal{A}$ as

$$\theta_t(a) = \frac{6\ln t}{\Delta(a)^2}, \tag{12}$$

and the random event

$$\mathcal{E}^t(e|S) = \{N_t(e|S) \geq \theta_t(e|S)\} \tag{13}$$

meaning that the arm $e|S$ is *sufficient sampled* at time $t$.

Lemma C.4 and Lemma C.5 is a technique adapted from [3], which can bound the error probability of the UCB policy for any sufficiently sampled arm.

**Lemma C.4.** *Consider two arms* $a_1$ *and* $a_2$ *that generate random variables on the support* $[0,1]$, *denoted as* $X_1$ *and* $X_2$, *respectively, and denote* $\mu_i = \mathbb{E}[X_i]$ *for* $i = 1, 2$. *Let* $\hat{X}_i$ *be the average of all i.i.d. samples from* $X_i$ *(i.e., the empirical mean of* $X_i$*), and* $N(a_i)$ *be the counter of samples. Denote* $\hat{X}_{i,N(a_i)}$ *be the average of the first* $N(a_i)$ *samples. W.L.O.G, assume* $d = \mu_1 - \mu_2 > 0$. *Define function* $\mathrm{rad}_t(a) := \sqrt{\frac{3\ln t}{2N(a)}}$. *For* $t = 1, 2, \ldots$, *if* $N(a_1), N(a_2) \geq 1$ *and* $N(a_2) > \frac{6\ln t}{d^2}$, *the following holds:*

$$\mathbb{P}\left[\hat{X}_{1,N(a_1)} + \mathrm{rad}_t(a_1) \leq \hat{X}_{2,N(a_2)} + \mathrm{rad}_t(a_2)\right]$$
$$\leq \mathbb{P}\left[\hat{X}_{1,N(a_1)} \geq \mu_1 - \mathrm{rad}_t(a_1)\right] + \mathbb{P}\left[\hat{X}_{2,N(a_2)} \leq \mu_2 + \mathrm{rad}_t(a_2)\right].$$

*Proof.* Since $N(a_1), N(a_2) \geq 1$ and $N(a_2) > \frac{6\ln t}{d^2}$, we can get $\mathrm{rad}_t(a_2) < \frac{d}{2}$ and $\mathrm{rad}_t(a_1) > 0$. Thus, $\mu_1 > \mu_2 + 2\mathrm{rad}_t(a_2)$. It is easy to verify that event $\hat{X}_{1,N(a_1)} + \mathrm{rad}_t(a_1) < \hat{X}_{2,N(a_2)} + \mathrm{rad}_t(a_2)$ implies at least one of the following events must hold:

$$\hat{X}_{1,N(a_1)} \leq \mu_1 - \mathrm{rad}_t(a_1) \tag{14}$$
$$\hat{X}_{2,N(a_2)} \geq \mu_2 + \mathrm{rad}_t(a_2). \tag{15}$$

Otherwise, assume that both of (14) and (15) are false, then $\hat{X}_{1,N(a_1)} + \mathrm{rad}_t(a_1) > \mu_1 > \mu_2 + 2\mathrm{rad}_t(a_2) > \hat{X}_{2,N(a_2)} + \mathrm{rad}_t(a_2)$, which causes a contradiction. Therefore, we can get $\mathbb{P}\left[\hat{X}_{1,N(a_1)} + \mathrm{rad}_t(a_1) < \hat{X}_{2,N(a_2)} + \mathrm{rad}_t(a_2)\right] = \mathbb{P}\left[(14) \text{ or } (15) \text{ is true}\right]$, and the lemma can be concluded by the union bound. $\qquad\square$

For any arm $a \in \mathcal{F}$, we denote the upper bound as

$$U_t(a) = \hat{X}(a) + \mathrm{rad}_t(N(a)). \tag{16}$$

As (3) defined in the main text, the counter $N_t(a) = \sum_{i=1}^{t-1} \mathbb{I}_i\{a\}$ is used to denoted the particular $N(a)$ at the beginning of the time step $t$, where $\mathbb{I}_i\{a\} \in \{0, 1\}$ indicates whether arm $a$ is updated at time $i$. (For OG-UCB, it is always updated once chosen.)

We need the following lemma, which bounds the expected value of the counter.

**Lemma C.5.** *For any time horizon* $T$ *and any* $k \geq 1$, *for any* $e \in \mathcal{N}_-(G_{k-1})$, *its counter satisfies*

$$\mathbb{E}\left[N_{T+1}(e|G_{k-1})\right] \leq \theta_T(e|G_{k-1}) + \frac{\pi^2}{3} + 1. \tag{17}$$

*Proof.* Consider a fixed prefix $G_{k-1}$ and an arm $e \in \mathcal{N}_-(G_{k-1})$. Suppose each arm in $\mathcal{N}(G_{k-1})$ is initialized (played once). Line 11 of Subroutine 2 (the UCB policy) indicates that, selecting element $e \in \mathcal{N}_-(G_{k-1})$ for phase $k$ implies that the random event $\{U_t(g_k|G_{k-1}) \leq U_t(e|G_{k-1})\}$ occurs. Then, we can bound the counter of playing $e|G_{k-1}$ by considering two disjoint random events, $\mathcal{E}^t(e|G_{k-1})$ and $\overline{\mathcal{E}^t(e|G_{k-1})}$ (sufficiently sampled and insufficiently sampled, respectively). That is

$$\mathbb{E}\left[N_{T+1}(e|G_{k-1})\right]$$

$$\leq \mathbb{E}\left[\sum_{t=1}^{T} \mathbb{I}\left\{\mathcal{O}_{0:k-1}^t \wedge U_t(g_k|G_{k-1}) \leq U_t(e|G_{k-1})\right\}\right]$$

$$= \mathbb{E}\left[\sum_{t=1}^{T} \mathbb{I}\left\{\mathcal{O}_{0:k-1}^t \wedge \overline{\mathcal{E}^t(e|G_{k-1})} \wedge U_t(g_k|G_{k-1}) \leq U_t(e|G_{k-1})\right\}\right]$$

$$+ \mathbb{E}\left[\sum_{t=1}^{T} \mathbb{I}\left\{\mathcal{O}_{0:k-1}^t \wedge \mathcal{E}^t(e|G_{k-1}) \wedge U_t(g_k|G_{k-1}) \leq U_t(e|G_{k-1})\right\}\right]$$

$$\leq \lceil\theta_T(e|G_{k-1})\rceil + \sum_{t=1}^{T} \mathbb{P}\left[\mathcal{O}_{0:k-1}^t \wedge \mathcal{E}^t(e|G_{k-1}) \wedge U_t(g_k|G_{k-1}) \leq U_t(e|G_{k-1})\right], \qquad (18)$$

where (18) follows from the definition of $\mathcal{E}^t(e|G_{k-1})$. Now we use Lemma C.4, in which arms are $a_1 = g_k|G_{k-1}$ and $a_2 = e|G_{k-1}$; empirical means $\hat{X}_1, \hat{X}_2$ are associated with $\hat{X}(g_k|G_{k-1})$ and $\hat{X}(e|G_{k-1})$, respectively; $\mu_1, \mu_2$ are used to denote their means; $N(a_1)$ and $N(a_2)$ are their counters. Then, for any $t$, we have that

$$\mathbb{P}\left[\bigcap_{i=1}^{k-1} \mathcal{O}_i^t \wedge \mathcal{E}^t(e|G_{k-1}) \wedge U_t(g_k|G_{k-1}) \leq U_t(e|G_{k-1})\right]$$

$$\leq \mathbb{P}\left[\mathcal{E}^t(e|G_{k-1}) \wedge U_t(g_k|G_{k-1}) \leq U_t(e|G_{k-1})\right]$$

$$\leq \sum_{N(a_1)=1}^{t} \mathbb{P}\left[\hat{X}_{1,N(a_1)} \geq \mu_1 - \text{rad}_t(a_1)\right] + \sum_{N(a_2)=\lceil\theta_t(a_2)\rceil}^{t} \mathbb{P}\left[\hat{X}_{2,N(a_2)} \leq \mu_2 + \text{rad}_t(a_2)\right]$$

$$\leq \sum_{N(a_1)=1}^{t} t^{-3} + \sum_{N(a_2)=1}^{t} t^{-3} = 2t^{-2}, \qquad (19)$$

where (19) holds because of Fact C.3. Summing over $t = 1, \ldots, T$ and using the convergence of Riemann zeta function (i.e., $\sum_{t=1}^{\infty} \frac{1}{t^2} = \frac{\pi^2}{6}$), the proof is completed. $\qquad\square$

**Theorem C.6** (Restatement of Theorem 3.1). *For any time $T$, Algorithm* OG-UCB *(Algorithm 1 + Subroutine 2) can achieve the greedy regret*

$$R^{\mathsf{G}}(T) \leq \sum_{a \in \Gamma_-(\sigma^{\mathsf{G}})} \left(\frac{6\Delta^*(a) \cdot \ln T}{\Delta(a)^2} + \left(\frac{\pi^2}{3} + 1\right)\Delta^*(a)\right),$$

*where $\sigma^{\mathsf{G}}$ is the greedy decision sequence.*

*Proof.* Now fix any $k$, and consider each term of the right-hand side in Lemma C.2, that is $\mathbb{E}\left[\sum_{t=1}^{T} \Delta(S^t) \cdot \mathbb{I}\left\{\mathcal{O}_{0:k-1}^t \cap \overline{\mathcal{O}_k^t}\right\}\right]$.

Suppose $\mathcal{O}_{0:k-1}^t \cap \overline{\mathcal{O}_k^t}$ happens, that is the algorithm selects $G_{k-1}$ at the first $k-1$ phases, and then picks $e \in \mathcal{N}_-(G_{k-1})$. By definition of the sunk-cost gap in (5), we can see that the regret at time $t$ is no more than $\Delta^*(e|G_{k-1})$. We we can get:

$$\mathbb{E}\left[\sum_{t=1}^{T} \Delta(S^t) \cdot \mathbb{I}\left\{\mathcal{O}_{0:k-1}^t \cap \overline{\mathcal{O}_k^t}\right\}\right]$$

$$\leq \mathbb{E}\left[\sum_{t=1}^{T}\Delta(S^t)\cdot\mathbb{I}\left\{\mathcal{O}_{0:k-1}^t\right\}\cdot\left(\sum_{e\in\mathcal{N}_-(G_{k-1})}\mathbb{I}_t\left\{e|G_{k-1}\right\}\right)\right] \tag{20}$$

$$\leq \sum_{e\in\mathcal{N}_-(G_{k-1})}\mathbb{E}\left[\sum_{t=1}^{T}\Delta(S^t)\cdot\mathbb{I}\left\{\mathcal{O}_{0:k-1}^t\right\}\cdot\mathbb{I}_t\left\{e|G_{k-1}\right\}\right]$$

$$\leq \sum_{e\in\mathcal{N}_-(G_{k-1})}\Delta^*(e|G_{k-1})\cdot\mathbb{E}\left[\sum_{t=1}^{T}\mathbb{I}\left\{\mathcal{O}_{0:k-1}^t\right\}\cdot\mathbb{I}_t\left\{e|G_{k-1}\right\}\right]$$

$$\leq \sum_{e\in\mathcal{N}_-(G_{k-1})}\Delta^*(e|G_{k-1})\cdot\mathbb{E}\left[N_{T+1}(e|G_{k-1})\right] \tag{21}$$

where (20) is by the union bound, and (21) is by the definition of $N_{T+1}(e|G_{k-1})$ and $\mathbb{I}\left\{\mathcal{O}_{0:k-1}^t\right\}\leq 1$. Thus, the greedy regret satisfies that

$$R^{\mathsf{G}}(T)\leq\sum_{k=1}^{m^{\mathsf{G}}}\mathbb{E}\left[\sum_{t=1}^{T}\Delta(S^t)\cdot\mathbb{I}\left\{\mathcal{O}_{1:k-1}^t\cap\overline{\mathcal{O}_k^t}\right\}\right] \tag{22}$$

$$\leq\sum_{k=1}^{m^{\mathsf{G}}}\sum_{e\in\mathcal{N}_-(G_{k-1})}\Delta^*(e|G_k)\cdot\mathbb{E}\left[N_T(e|G_{k-1})\right] \tag{23}$$

$$=\sum_{a\in\Gamma_-(\sigma^{\mathsf{G}})}\Delta^*(a)\cdot\mathbb{E}\left[N_T(a)\right] \tag{24}$$

$$\leq\sum_{a\in\Gamma_-(\sigma^{\mathsf{G}})}\left(\frac{6\Delta^*(a)\cdot\ln T}{\Delta(a)^2}+\left(\frac{\pi^2}{3}+1\right)\Delta^*(a)\right), \tag{25}$$

where (22) is derived from Lemma C.2; (23) is from (21); (24) is due to Definition 3.2; and (25) follows Lemma C.5. Therefore, the theorem is concluded. $\qquad\square$

# D Analysis for OG-LUCB$_{\epsilon,\delta}$ and OG-LUCB-R in Section 4

In this section, we provide the analysis of Algorithm OG-LUCB$_{\epsilon,\delta}$ (Algorithm 1 + Subroutine 3) and Algorithm OG-LUCB-R (Algorithm 4). In the following, we first derive the probability inequality of confidence events. Second, we analyze the exploration time in Appendix D.1. Assuming that the total time horizon $T$ is known, we provide an $O(\log T)$ $\epsilon$-quasi greedy regret in Appendix D.2. Finally, we show that we can remove the assumption of knowing $T$ and still obtain an $O(\log T)$ regret bound (by Algorithm OG-LUCB-R).

Given any arm $a=e|S\in\mathcal{A}$, define event $\mathcal{C}^t(e|S)=\left\{\left|\hat{X}_t(e|S)-X(e|S)\right|<\mathrm{rad}_t(e|S)\right\}$ (confidence bound holds). For any prefix $S$, define $\mathcal{C}^t(S)=\bigcup_{e\in\mathcal{N}(S)}\mathcal{C}^t(e|S)$ for all the arms $e|S$.

Recall that for any $S$, $e\in\mathcal{N}(S)$, by definition

$$\Delta(e|S_k)=\begin{cases}\overline{f}(g_S^*|S)-\overline{f}(e|S), & e\neq g_S^*\\ \overline{f}(e|S)-\max_{a\in\mathcal{N}(S)\setminus\{g_S^*\}}\overline{f}(a|S), & e=g_S^*\end{cases},$$

and the hardness coefficient with the prefix $S$ as

$$H_S^{\epsilon}:=\sum_{e\in\mathcal{N}(S)}\frac{1}{\max\left\{\Delta(e|S)^2,\epsilon^2\right\}}.$$

**Lemma D.1.** *Fix any* $\delta\in(0,1)$. *Suppose* $\mathrm{rad}_t(s)=\sqrt{\frac{\ln(4Wt^3/\delta)}{2N_t(s)}}$. *For any prefix* $S$, $\bigcup_{t=1}^{\infty}\mathcal{C}_t(S)$ *holds with probability at least* $1-\delta$.

*Proof.* From Fact C.3, we know that for any $e\in\mathcal{N}(S)$,

$$\mathbb{P}\left[\mathcal{C}^t(e|S)\right]=\mathbb{P}\left[\left|\hat{X}_t(e|S)-X(e|S)\right|<\mathrm{rad}_t(e|S),N_t(e|S)=1,\cdots,t\right]$$

$$\geq 1 - 2 \cdot t \cdot \frac{\delta}{4Wt^3} = 1 - \frac{\delta}{2Wt^2}.$$

Then, by summing all time steps $t$ and all elements $e \in \mathcal{N}(S)$ ($|\mathcal{N}(S)| \leq W$), we can conclude that

$$\mathbb{P}\left[\bigcup_{t=1}^{\infty} \mathcal{C}^t(S)\right] \geq 1 - \sum_{t=1}^{\infty} \sum_{e \in \mathcal{N}(S)} \frac{\delta}{2Wt^2} \geq 1 - \sum_{t=1}^{\infty} \frac{\delta}{2t^2}$$
$$\geq 1 - \frac{\pi^2 \delta}{12} \tag{26}$$
$$\geq 1 - \delta,$$

where (26) is by Riemann zeta function. $\qquad\square$

### D.1 Exploration time for OG with a first-explore-then-exploit policy (OG-LUCB$_{\epsilon,\delta}$)

Since we use a specialized version of CLUCB-PAC in [9] as MaxOracle to explore the top-one element in the support of $[0,1]$ for each phase, and henceforth our analysis starts from the result of sample complexity of CLUCB-PAC by setting $R = \frac{1}{2}$ and $\mathrm{width}(\mathcal{M}) = 2$.

In Algorithm OG-LUCB$_{\epsilon,\delta}$, for any arm $a = e|S \in \mathcal{A}$, denote $N_t^{\mathsf{E}}(a)$ as the counter of playing arm $a$ during the exploration stage at the beginning of time step $t$. The algorithm turns from the exploration to the exploitation some time and the counter will not change, therefore we may use $N_\infty^{\mathsf{E}}(a)$ to obtain its final value. For any prefix $S$, denote arms $A = \{e|S : \forall e \in \mathcal{N}(S)\}$. Denote $t_0(S) := \sum_{e|S \in A} N_{t_0}^{\mathsf{E}}(e|S)$ as the total exploration time of all elements in $\mathcal{N}(S)$, such that for any time $t \geq t_0(S)$, $I^{\mathsf{exploit}}(A)$ is initialized.

Notice that we use CLUCB-PAC for each phase, therefore the following lemma can be adapted from the intermediate step in proving Theorem 5 of [9].

**Lemma D.2.** *For any phase $k \geq 0$, fix prefix $S_k$. Suppose $\epsilon \geq 0$, $\delta \in (0,1)$, and $\bigcup_{t=1}^{\infty} \mathcal{C}^t(S_k)$ holds. If it goes to the exploitation stage (Line 12 of Subroutine 3) with $I^{\mathsf{exploit}}(A) = s^*$ at time $t_0 = t_0(S_k)$, then:*

1. $f(s^*|S_k) \geq f(g_{S_k}^*|S_k) - \epsilon$;

2. *In addition, for any $s \in \mathcal{N}(S_k)$,*

$$N_{t_0}^{\mathsf{E}}(s|S_k) \leq \min\left\{\frac{18}{\Delta(s|S_k)^2}, \frac{16}{\epsilon^2}\right\} \ln\left(4Wt_0^3/\delta\right) + 1. \tag{27}$$

Notice that event $\bigcup_{t=1}^{\infty} \mathcal{C}^t(S_k)$ holds with probability at least $1 - \delta$, which is guaranteed by Lemma D.1. From the above lemma, we may further derive the following bound for $t_0(S_k)$.

**Lemma D.3.** *With the same setting as Lemma D.2. For $t_0 = t_0(S_k)$, denote $t_c = t_c(S_k) := 499H_{S_k}^{\epsilon} \ln(4WH_{S_k}^{\epsilon}/\delta) + 2W$ then we can get:*

1. $t_0 \leq t_c$.

2. $\ln\left(4Wt_c^3/\delta\right) \leq 7\ln\left(1996WH_{S_k}^{\epsilon}/\delta\right)$.

*Proof.* **Property (1):** First of all, (1) holds trivially if $W \geq \frac{t_0}{2}$, thus we only need to show the case $W < \frac{t_0}{2}$.

Since $t_0 = \sum_{e \in \mathcal{N}(S_k)} N_{t_0}^{\mathsf{E}}(e|S_k)$, it can be implied from Lemma D.2 that, with probability at least $1 - \delta$,

$$t_0 \leq 18H_{S_k}^{\epsilon} \ln(4Wt_0^3/\delta) + W. \tag{28}$$

We assume $t_0 = CH_{S_k}^{\epsilon} \ln(4WH_{S_k}^{\epsilon}/\delta) + W$, for some constant $C > 0$. When $W < \frac{t_0}{2}$, $t_0 < 2CH_{S_k}^{\epsilon} \ln(4WH_{S_k}^{\epsilon}/\delta)$. Then, rewrite (28) as

$$t_0 \leq W + 18H_{S_k}^{\epsilon} \ln(4W/\delta) + 54H_{S_k}^{\epsilon} \ln(t_0)$$

$$\begin{aligned}
&< W + 18H^\epsilon_{S_k}\ln(4W/\delta) + 54H^\epsilon_{S_k}\ln(2CH^\epsilon_{S_k}\ln(4WH^\epsilon_{S_k}/\delta)) \\
&\leq W + 18H^\epsilon_{S_k}\ln(4W/\delta) + 54H^\epsilon_{S_k}\left(\ln(2C)+\ln(H^\epsilon_{S_k})\right) + 54H^\epsilon_{S_k}\ln(4WH^\epsilon_{S_k}/\delta) \\
&\leq W + 72H^\epsilon_{S_k}\ln(4WH^\epsilon_{S_k}/\delta) + 54H^\epsilon_{S_k}\left(\ln(2C)\ln(4WH^\epsilon_{S_k}/\delta)+\ln(4WH^\epsilon_{S_k}/\delta)\right) \\
&\leq W + (126 + 54\ln 2C)\,H^\epsilon_{S_k}\ln(4WH^\epsilon_{S_k}/\delta).
\end{aligned} \tag{29}$$

Solve $126 + 54\ln 2C < C$, and we can get the minimum integer solution $C = 499$. When $C \geq 499$, from (29), $t_0 < W + CH^\epsilon_{S_k}\ln(4WH^\epsilon_{S_k}/\delta) = t_0$, which cause a contradiction. Thus, we can conclude that $t_0 \leq 499H^\epsilon_{S_k}\ln(4WH^\epsilon_{S_k}/\delta) + 2W$.

**Property (2):** We can simplify $\ln\left(4Wt_c^3/\delta\right)$ in (27) as follows.

From Property 1, since $\ln(a+b) \leq \ln(a) + \ln(b)$ for all $a, b \geq e$, we have

$$\begin{aligned}
\ln(t_c) &\leq \ln\left(499H^\epsilon_{S_k}\ln(4WH^\epsilon_{S_k}/\delta)\right) + \ln(4W) \\
&\leq \ln(499H^\epsilon_{S_k}) + \ln\ln(4WH^\epsilon_{S_k}/\delta) + \ln(4W) \\
&\leq \ln(499H^\epsilon_{S_k}) + 2\ln(4W) + \ln\left(H^\epsilon_{S_k}\right) + \ln(1/\delta),
\end{aligned} \tag{30}$$

then

$$\begin{aligned}
\ln\left(4Wt_c^3/\delta\right) &\leq \ln(4W) + \ln(1/\delta) + 3\ln(t_c) \\
&\leq 7\ln(4W) + 4\ln(1/\delta) + 6\ln\left(499H^\epsilon_{S_k}\right) \\
&\leq 7\ln\left(1996WH^\epsilon_{S_k}/\delta\right),
\end{aligned}$$

which ends the proof. $\qquad\square$

Define random event iteratively, for any $k \geq 1$, given prefix $S_{k-1}$,

$$\mathcal{Q}_k(S_{k-1}) = \left\{\overline{f}(s_k|S_{k-1}) \geq \overline{f}(g^*_{S_{k-1}}|S_{k-1}) - \epsilon\right\}, \tag{31}$$

where $s_k = I^{\mathsf{exploit}}(A)$ and $A = \{e|S_{k-1} : \forall e \in \mathcal{N}(S_{k-1})\}$.

**Theorem D.4** (Restatement of Theorem 4.1). *Given any $\epsilon \geq 0$ and $\delta \in (0,1)$, suppose after the total exploration time $T^{\mathsf{E}} = T^{\mathsf{E}}(\delta)$, Algorithm $\mathsf{OG\text{-}LUCB}_{\epsilon,\delta}$ sticks to a stable sequence $\sigma = \langle S_0, S_1, \cdots, S_{m'}\rangle$. With probability at least $1 - m\delta$ ($m$ is the largest length of a feasible set), the following three claims hold:*

1. *$\sigma$ is an $\epsilon$-quasi greedy sequence;*

2. *Let $t_c(S) := 499H^\epsilon_S\ln(4WH^\epsilon_S/\delta) + 2W$. For any arm $e|S$ in the decision frontier $\Gamma(\sigma)$,*

$$N^{\mathsf{E}}_\infty(e|S) \leq \min\left\{\frac{18}{\Delta(e|S)^2}, \frac{16}{\epsilon^2}\right\}\ln\left(4Wt_c^3(S)/\delta\right) + 1. \tag{32}$$

3. *The total exploration time $T^{\mathsf{E}}$ satisfies that*

$$T^{\mathsf{E}} \leq \sum_{e|S\in\Gamma(\sigma)} N^{\mathsf{E}}_\infty(e|S) \leq 127\sum_{k=0}^{m'-1} H^\epsilon_S\ln\left(1996WH^\epsilon_S/\delta\right). \tag{33}$$

*Proof.* **Property (1):** Similar to the set decomposition in Lemma C.1, by De Morgan's laws, we know that

$$\overline{\bigcap_{i=1}^{m'}\mathcal{Q}_i^t(S_{i-1})} = \left(\overline{\mathcal{Q}_1^t(S_0)}\right) \cup \left(\mathcal{Q}_1^t(S_0)\cap\overline{\mathcal{Q}_2^t(S_1)}\right) \cup \cdots \cup \left(\left(\bigcap_{i=1}^{m'-1}\mathcal{Q}_i^t(S_{i-1})\right)\cap\overline{\mathcal{Q}_k^t(S_{m'-1})}\right).$$

By Lemma D.2, we know that for $k = 1, 2, \cdots, m'$, the $k$-th term above satisfies

$$\mathbb{P}\left[\left(\bigcap_{i=1}^{k-1}\mathcal{Q}_i^t(S_{i-1})\right)\cap\overline{\mathcal{Q}_k^t(S_{k-1})}\right] \leq \mathbb{P}\left[\overline{\mathcal{Q}_k^t(S_{k-1})}\right] \leq \delta. \tag{34}$$

Therefore, by union bound and $m' \le m$, we have that

$$\mathbb{P}\left[\bigcap_{i=1}^{m'} \mathcal{Q}_i^t(S_{k-1})\right] = 1 - \mathbb{P}\left[\overline{\bigcap_{i=1}^{m'} \mathcal{Q}_i^t(S_{k-1})}\right] \le 1 - \sum_{k=1}^{m'} \delta \le 1 - m\delta,$$

and we prove that $\sigma$ is an $\epsilon$-quasi greedy sequence.

**Property (2):** Assume that $\bigcap_{i=1}^{m'} \mathcal{Q}_i^t(S_{i-1})$ holds. Since we denote $t_c(S) = 499 H_S^\epsilon \ln(4WH_S^\epsilon/\delta) + 2W$, applying Property 1 of Lemma D.3, we can get $t_0(S_{k-1}) \le t_c(S_{k-1})$. Then, use Property 2 of Lemma D.2, and it follows immediately that

$$N_\infty^{\mathsf{E}}(e|S_{k-1}) \le \min\left\{\frac{18}{\Delta(e|S_{k-1})^2}, \frac{16}{\epsilon^2}\right\} \ln\left(4Wt_c^3(S_{k-1})/\delta\right) + 1, \tag{35}$$

where $e|S_{k-1}$ is any arm occurred in the decision frontier $\Gamma(\sigma)$.

**Property (3):** It is easy to see that the total exploration time $T^{\mathsf{E}} \le \sum_{e|S \in \Gamma(\sigma)} N_\infty^{\mathsf{E}}(e|S)$. Sum up each phase $k = 0, 1, \cdots, m' - 1$ and each element $e \in \mathcal{N}(S_k)$, and use $\ln\left(4Wt_c^3(S_{k-1})\right) \le 7 \ln\left(1996 WH_{S_k}^\epsilon/\delta\right)$ by Property 2 of Lemma D.3. Therefore we can get:

$$\begin{aligned}
T^{\mathsf{E}} \le \sum_{e|S \in \Gamma(\sigma)} N_\infty^{\mathsf{E}}(e|S) &\le \sum_{k=0}^{m'-1} \sum_{e \in \mathcal{N}(S_k)} N_\infty^{\mathsf{E}}(e|S_k) \\
&\le \sum_{k=0}^{m'-1} \left(126 H_{S_k}^\epsilon \ln\left(1996 WH_{S_k}^\epsilon/\delta\right) + |\mathcal{N}(S_k)|\right) \\
&\le 127 \sum_{k=0}^{m'-1} H_{S_k}^\epsilon \ln\left(1996 WH_{S_k}^\epsilon/\delta\right). \tag{36}
\end{aligned}$$

$\square$

## D.2  Time Horizon $T$ is Known

We first assume that we know the total time horizon $T$. In this case, we may let $\delta = \frac{1}{T}$ in OG-LUCB$_{\epsilon,\delta}$, and we can obtain the following theorem.

**Theorem D.5** (Restatement of Theorem 4.2). *Given any $\epsilon \ge 0$. When total time $T$ is known, in Algorithm* OG-LUCB$_{\epsilon,\delta}$*, we associate $\delta = \frac{1}{T}$. Suppose $\sigma = \langle S_0, S_1, \cdots, S_{m'}\rangle$ is the sequence selected by Algorithm* OG-LUCB$_{\epsilon,\delta}$ *at time $T$, and define function*

$$R^{\mathsf{Q},\sigma}(T) := \sum_{e|S \in \Gamma(\sigma)} \Delta^*(e|S) \min\left\{\frac{127}{\Delta(e|S)^2}, \frac{113}{\epsilon^2}\right\} \ln\left(1996 WH_S^\epsilon T\right) + \Delta_{\max} m, \tag{37}$$

*where $m$ is the largest length of a feasible set and $H_S^\epsilon$ is defined in (7). Then, the $\epsilon$-quasi regret satisfies*

$$R^{\mathsf{Q}}(T) \le R^{\mathsf{Q},\sigma}(T) = O\left(\frac{Wm\Delta_{\max}}{\max\{\Delta^2, \epsilon^2\}} \log T\right), \tag{38}$$

*where $\Delta$ is the minimum unit gap.*

*Proof.* First, we claim that

$$R^{\mathsf{Q}}(T) \le \sum_{e|S \in \Gamma(\sigma)} \Delta^*(e|S)\left(\min\left\{\frac{18}{\Delta(e|S)^2}, \frac{16}{\epsilon^2}\right\} \ln\left(4Wt_{c,T}^3(S) \cdot T\right) + 1\right) + \Delta_{\max} m, \tag{39}$$

where $t_{c,T}(S) = 499 H_S^\epsilon \ln(4WH_S^\epsilon \cdot T) + 2W$ for any $S$.

From Property 1 of Theorem D.4 , we know that with probability $1 - \frac{m}{T}$, the sequence $\sigma$ is a stable $\epsilon$-greedy sequence, i.e., all confidence events $\bigcap_{i=1}^{m'} \mathcal{Q}_i^t(S_{k-1})$ hold.

We may sum up exploration numbers of the arms ($N_\infty^{\mathsf{E}}(\cdot)$'s) in Property 2 of Theorem D.4, so that the first term of (39) can be easily derived. For the second term of (39), it is because $\mathsf{OG\text{-}LUCB}_{\epsilon,\delta}$ pays at most regret $\Delta_{\max}T$ with probability $\frac{m}{T}$, when any confidence events fails, which contributes at most $\Delta_{\max}m$ to the regret.

Then, apply Property 2 of Lemma D.3 and $\delta = \frac{1}{T}$, i.e., $\ln\left(4Wt_{c,T}^3(S) \cdot T\right) \leq 7\ln\left(1996WH_{S_k}^\epsilon/\delta\right)$, and henceforth it is obvious to see that

$$R^{\mathsf{Q}}(T) \leq R^{\mathsf{Q},\sigma}(T) = \sum_{e|S\in\Gamma(\sigma)} \Delta^*(e|S) \min\left\{\frac{127}{\Delta(e|S)^2}, \frac{113}{\epsilon^2}\right\} \ln\left(1996WH_S^\epsilon T\right) + \Delta_{\max}m.$$

Furthermore, if we replace $\Delta^*(e|S)$ with the maximum gap $\Delta_{\max}$, and $\Delta(e|S)$ with the minimum unit gap $\Delta$, it is easily derived that $R^{\mathsf{Q}}(T) \leq R^{\mathsf{Q},\sigma}(T) = O(\frac{Wm\Delta_{\max}}{\max\{\Delta^2,\epsilon^2\}} \log T)$, where $W$ is the largest width and $m$ is the largest length. $\qquad\square$

Notice that, from the above theorem, the $\epsilon$-quasi regret is bounded within $O(\log T)$ for any time horizon $T$, even though $T$ may be less than the total exploration time $T^{\mathsf{E}} = T^{\mathsf{E}}\left(\frac{1}{T}\right)$. It is because $N_\infty^{\mathsf{E}}(\cdot)$ is the upper bound approaching the infinite time, and the regret is a non-decreasing function with the time horizon $T$. Therefore, the above regret is satisfied for $T < T^{\mathsf{E}}\left(\frac{1}{T}\right)$ as well.

### D.3 Time Horizon $T$ is not Known

When the time horizon $T$ is not known, we use Algorithm $\mathsf{OG\text{-}LUCB\text{-}R}$ (Algorithm 4) which restarts the internal $\mathsf{OG\text{-}LUCB}_{\epsilon,\delta}$ for different epochs. The following lemma is useful in proving Theorem D.7.

**Lemma D.6.** *For any $i = 1, 2, \cdots, k$, for any $c_i, b_i > 0$ and $\phi_i = e^{2^i}$,*

$$\sum_{i=1}^{k}(c_i \cdot \ln(\phi_i) + b_i) \leq 4 \cdot \left(\max_{i=1,\cdots,k} c_i\right) \cdot \ln(\phi_{k-1}) + k \cdot \left(\max_{i=1,\cdots,k} b_i\right).$$

*Proof.* Since $\phi_i = e^{2^i}$, then $\ln(\phi_i) = 2^i$, an we can get

$$\sum_{i=1}^{k} \ln(\phi_i) = 2^1 + 2^2 + \cdots + 2^k < 4 \times 2^{k-1} = 4\ln(\phi_{k-1}).$$

Therefore, it follows immediately that

$$\sum_{i=1}^{k}(c_i \cdot \ln(\phi_i) + b_i) \leq \left(\max_{i=1,\cdots,k} c_i\right) \sum_{i=1}^{k} \ln(\phi_i) + k \cdot \left(\max_{i=1,\cdots,k} b_i\right)$$

$$\leq 4 \cdot \left(\max_{i=1,\cdots,k} c_i\right) \cdot \ln(\phi_{k-1}) + k \cdot \left(\max_{i=1,\cdots,k} b_i\right).$$

$\qquad\square$

**Theorem D.7** (Restatement of Theorem 4.3). *Given any $\epsilon \geq 0$. Use $\phi_\ell$ and $\tau_\ell$ defined in (8), and function $R^{\mathsf{Q},\sigma}(T)$ defined Theorem 4.2. In Algorithm $\mathsf{OG\text{-}LUCB\text{-}R}$, suppose $\sigma^{(\ell)} = \langle S_0^{(\ell)}, S_1^{(\ell)}, \cdots, S_{m^{(\ell)}}^{(\ell)}\rangle$ is the sequence selected by the end of $\ell$-th epoch of $\mathsf{OG\text{-}LUCB}_{\epsilon,\delta}$, where $m^{(\ell)}$ is its length. For any time $T$, denote the final epoch as $K = K(T)$ such that $\tau_{K-1} < T \leq \tau_K$, and the $\epsilon$-quasi greedy regret satisfies that*

$$R^{\mathsf{Q}}(T) \leq \sum_{\ell=1}^{K} R^{\mathsf{Q},\sigma^{(\ell)}}(\phi_\ell). \tag{40}$$

*Furthermore, we can get*

$$R^{\mathsf{Q}}(T) \leq 4 \cdot \left( \max_{\ell=1,\cdots,K} c_\ell \right) \cdot \ln(T) + \left( \max_{\ell=1,\cdots,K} b_\ell \right) \cdot \log_2\left(2\ln T\right) \tag{41}$$

$$= O\left( \frac{Wm\Delta_{\max}}{\max\{\Delta^2, \epsilon^2\}} \log T \right), \tag{42}$$

*where*

$$c_\ell := \sum_{e|S\in\Gamma(\sigma^{(\ell)})} \Delta^*(e|S) \min\left\{ \frac{127}{\Delta\left(e|S\right)^2}, \frac{113}{\epsilon^2} \right\},$$

$$b_\ell := \sum_{e|S\in\Gamma(\sigma^{(\ell)})} \Delta^*(e|S) \min\left\{ \frac{127}{\Delta\left(e|S\right)^2}, \frac{113}{\epsilon^2} \right\} \ln\left(1996WH_S^\epsilon\right) + \Delta_{\max}m,$$

*and $\Delta$ is the minimum unit gap.*

*Proof.* Algorithm OG-LUCB-R restarts the internal OG-LUCB$_{\epsilon,\delta}$ for each epoch. Since $\epsilon$-quasi regret for epoch $\ell$ during time $(\tau_{\ell-1}, \tau_\ell]$ is no more than $R^{\mathsf{Q},\sigma^{(\ell)}}(\phi_\ell)$, then the $\epsilon$-quasi greedy regret in (40) follows naturally by accumulating each interval.

For any $\ell$, $\phi_{\ell-1} \leq \tau_{\ell-1}, \tau_\ell = \sum_{i=1}^\ell e^{2^i} < 2e^{2^\ell} = 2\phi_\ell$. Since $\tau_{K-1} < T \leq \tau_K$, we have $\phi_{K-1} < T < 2\phi_K$ and $K < \log_2(2\ln T)$.

Furthermore, we can get

$$\sum_{\ell=1}^K R^{\mathsf{Q},\sigma^{(\ell)}}(\phi_\ell) \leq \sum_{\ell=1}^K \left( c_\ell \ln\left(\phi_i\right) + b_\ell \right) \tag{43}$$

$$\leq 4 \cdot \left( \max_{\ell=1,\cdots,K} c_\ell \right) \cdot \ln(\phi_{K-1}) + \left( \max_{\ell=1,\cdots,K} b_\ell \right) \cdot K \tag{44}$$

$$\leq 4 \cdot \left( \max_{\ell=1,\cdots,K} c_\ell \right) \cdot \ln(T) + \left( \max_{\ell=1,\cdots,K} b_\ell \right) \cdot \log_2\left(2\ln T\right), \tag{45}$$

where (43) is by definition of $c_i$ and $b_i$, and the form of $R^{\mathsf{Q},\sigma^{(\ell)}}$ defined (37); (44) is from Lemma D.6; and (45) is because $\phi_{K-1} < T$ and $K < \log_2(2\ln T)$. Use the similar technique in the proof of Theorem D.5, and we know that (45) is also in $O\left( \frac{Wm\Delta_{\max}}{\max\{\Delta^2,\epsilon^2\}} \log T \right)$. Therefore, the theorem is concluded. □

# E   Proof of Lower Bound in Section 5

In this section, we construct an instance of the prize-collecting problem, and provide its theoretical analysis. We first recall the *prize-collecting problem* defined in the main text as follows.

**Problem.** Consider $m$ bandits, each of which has $W$ elements. For each bandit $i = 1, 2, \ldots, m$, denote set $E_i = \{e_{i,1}, e_{i,2}, \ldots, e_{i,W}\}$. In this problem, the player needs to select one element from each bandit in order. The accessible set system is defined as $(E, \mathcal{F})$, where $E = \bigcup_{i=1}^m E_i$, $\mathcal{F} = \cup_{i=1}^m \mathcal{F}_i \cup \{\emptyset\}$, and $\mathcal{F}_i = \{S \subseteq E : |S| = i, \forall k : 1 \leq k \leq i, |S \cap E_k| = 1\}$. The reward function $f : \mathcal{F} \times \Omega \to [0, m]$ is non-decreasing in the first parameter, where the form of $f$ is unknown to the player. Let the minimum unit gap be

$$\Delta := \min\left\{ \overline{f}(g_S^*|S) - \overline{f}(e|S) : \forall S \in \mathcal{F}, \forall e \in \mathcal{N}_-(S) \right\} > 0, \tag{46}$$

where the value of $\Delta$ is also unknown to the player. The objective of the player is to minimize the greedy regret.

**Instance $\mathcal{P}$.** We construct a problem instance $\mathcal{P}$ as follows.

We arbitrarily pick the greedy decision sequence $\sigma^{\mathsf{G}} = \langle G_0, G_1, \ldots, G_m \rangle \in \mathcal{F}^{m+1}$ where $G_0 = \emptyset$ and $G_i = G_{i-1} \cup \{e_i^{\mathsf{G}}\}$ for each $i$.

Assume that $0 < \mu_1 < \mu_2 < \mu_3 < 1$, and $\Delta := \mu_2 - \mu_1 > 0$. Consider that the environment's randomness comes from Bernoulli random variables $\omega_t = \left(\omega_{t,1}^1, \omega_{t,2}^1, \cdots, \omega_{t,1}^m, \omega_{t,2}^m\right) \in \{0,1\}^{2m} = \Omega$, with $\mathbb{E}[\omega_{t,1}^i] = \mu_1$ (low prize) for any $i = 1, 2, \cdots, m$; $\mathbb{E}[\omega_{t,2}^i] = \mu_2$ (medium prize) for $i = 1, 2, \cdots, m-1$, and $\mathbb{E}[\omega_{t,2}^m] = \mu_3$ (high prize). For convenience, given a feasible set $S$, we define indicator $\mathbb{I}_i^{\mathsf{G}} := \mathbb{I}\{e_i^{\mathsf{G}} \in S\}$ for each $i$. The exact form of reward function is $f(S, \omega_t) := \sum_{i=1}^m f^{(i)}(S, \omega_t)$, where for each $i = 1, 2, \cdots, m$,

$$f^{(i)}(S, \omega_t) = \begin{cases} \omega_{t,1}^i \overline{\mathbb{I}_i^{\mathsf{G}}} + \omega_{t,2}^i \mathbb{I}_i^{\mathsf{G}}, & G_{i-1} \subseteq S \\ \omega_{t,1}^i, & \text{otherwise} \end{cases}. \tag{47}$$

It is only accessed as a value oracle $f_t(S) := f(S, \omega_t)$ with a given feasible set $S$, and *the player does not know the form of the reward function*. For example, at time $t$, provided that we have already selected the greedy prefix $G_{i-1}$ for the first $i-1$ phases (that is $\cup_{j=1}^{i-1}\{e_i^{\mathsf{G}}\} \subseteq S$), if we choose the greedy element $e_i^{\mathsf{G}}$ at phase $i$ (that is $\mathbb{I}_i^{\mathsf{G}} = 1$), then the marginal reward $f^{(i)}(S, \omega_t) = \omega_{t,2}^i$, and the feedback for arm $e_i^{\mathsf{G}}|G_{i-1}$ is $f_t(e_i^{\mathsf{G}}|G_{i-1}) = f^{(i)}(S, \omega_t) = \omega_{t,2}^i$; otherwise, we will get $\omega_{t,1}^i$ for choosing the sub-optimal element. Moreover, only if all elements along the greedy sequence are chosen, that is $S = G$, can we get the marginal reward $f^{(m)}(S, \omega_t) = \omega_{t,2}^m$ for the last phase, where $\mathbb{E}[\omega_{t,2}^m] = \mu_3$.

Collecting prizes means that the player should find the greedy sequence to gain medium prizes ($\mathbb{E}[\omega_{t,2}^i] = \mu_2 > \mu_1 = \mathbb{E}[\omega_{t,1}^i]$) for the first $m-1$ phases and achieve the high prize ($\mathbb{E}[\omega_{t,2}^m] = \mu_3$) for the last phase. Since the maximal decision sequence is $m$ phases, it is easy to infer that, the minimum reward and maximum reward in expectation are $m\mu_1$ and $(m-1)\mu_2 + \mu_3$ respectively, and henceforth the maximum gap is $\Delta_{\max} = m\Delta + (\mu_3 - \mu_2)$. When the player mistakenly chooses a wrong element for some phase, denote the minimum gap as $\Delta_{\min} = \Delta + (\mu_3 - \mu_2)$, and $\Delta_{\min}$ is the minimum penalty incurred.

Denote greedy arms as $\mathcal{A}^{\mathsf{G}} = \{e_i^{\mathsf{G}}|G_{i-1} : \forall i = 1, 2, \cdots, W\}$. We say an algorithm is *consistent*, if the sum of playing all arms $a \in \mathcal{A} \setminus \mathcal{A}^{\mathsf{G}}$ is in $o(T^\eta)$, for any $\eta > 0$, i.e., $\mathbb{E}[\sum_{a \in \mathcal{A} \setminus \mathcal{A}^{\mathsf{G}}} N_T(a)] = o(T^\eta)$.

## E.1 Lower Bound $\Omega\left(\frac{mW\Delta_{\max}}{\Delta^2} \log T\right)$

In this subsection, we set $\mu_3 - \mu_2$ to be a constant, and derive the $\Omega\left(\frac{mW\Delta_{\min}}{\Delta^2} \log T\right)$ greedy regret lower bound.

Fix any $\ell = 1, 2, \cdots, m-1$. As is illustrated in Figure 1, there exists some feasible sequence $\sigma$ that coincides with the greedy sequence $\sigma^{\mathsf{G}}$ for the first $\ell$ arms, and deviates from it for the next $d := m - \ell$ arms, that is $\sigma = \langle G_0, G_1, \cdots, G_\ell, S_{\ell+1}, \cdots, S_m \rangle$, and $S_{\ell+1} \neq G_\ell \cup \{e_i^{\mathsf{G}}\}$. For convenience, we refer those $d$ arms as $a_1, a_2, \ldots, a_d$, and for any fixed time $T \geq T_0$ ($T_0$ is sufficiently large), we use $\vec{N} = (N(a_1), N(a_2), \cdots, N(a_d))$ to denote counters of $a_1, \ldots, a_d$, respectively, where each $N(a_k) := N_T(a_k)$. Obviously, $N(a_1) \geq N(a_2) \geq \cdots \geq N(a_d)$. Among all those sequences, since we have $W$ options for each arm $a_k$ after $a_{k-1}$, by Pigeonhole principle, there exists one sequence satisfying that

$$N(a_2) \leq \frac{1}{W} N(a_1), \ N(a_3) \leq \frac{1}{W} N(a_2), \ \cdots, \ \text{and } N(a_d) \leq \frac{1}{W} N(a_{d-1}). \tag{48}$$

W.L.O.G., assume that $\sigma$ satisfies condition (48).

In Lemma E.1, we will show that when $d$ is large enough, the number of playing $a_1$ is $\Omega\left(\frac{1}{(\mu_2 - \mu_1)^2} \log T\right)$.

**Lemma E.1.** *Assume that $\frac{1}{4} < \mu_1 < \mu_2 = \frac{1}{2} < \mu_3 = \frac{3}{4}$. Suppose $W \geq 2$, and integer $d$ satisfies that $d \geq \log_W \left(\frac{2W}{3(\mu_2 - \mu_1)^2}\right)$, (i.e., $(\mu_2 - \mu_1)^2 \geq \frac{2}{3W^{d-1}}$). For any consistent algorithm, the number*

Figure 1: Illustration of the sequence $\sigma$.

$d$ arms in $\sigma$ are denoted as $a_1, a_2, ..., a_d$, respectively.

*of playing arm $a_1$ in the sequence $\sigma$ satisfies that*

$$\lim_{t \to +\infty} \frac{\mathbb{E}[N_t(a_1)]}{\ln t} \geq \frac{0.39}{\mathrm{KL}(\mu_1 || \mu_2)},$$

*where $\mathrm{KL}(p||q) = p \ln \frac{p}{q} + (1-p) \ln \frac{1-p}{1-q}$ is Kullback-Leibler divergence of Bernoulli random variables.*

*Proof.* We keep all arms except $a_1, a_2, \ldots, a_d$ by default, and perturb those arms to construct the following two hypotheses:

$$\text{Null hypothesis } H_\sigma^0 : \qquad \overline{f}(a_i) = \mu_1, i \in \{1, \cdots, d-1\}, \quad \overline{f}(a_d) = \mu_1; \qquad (49)$$

$$\text{Alternative hypothesis } H_\sigma' : \qquad \overline{f}(a_i) = \lambda, i \in \{1, \cdots, d-1\}, \quad \overline{f}(a_d) = \mu_3, \qquad (50)$$

where $\lambda \in (\mu_2, 1]$ is some constant to be determined later (in (51)). Note that $\sigma^{\mathsf{G}}$ is its greedy sequence in $H_\sigma^0$, while $\sigma$ is the greedy sequence in $H_\sigma'$. For simplicity, we use $\mathbb{P}[\cdot]$ and $\mathbb{P}'[\cdot]$, $\mathbb{E}[\cdot]$ and $\mathbb{E}'[\cdot]$ to denote the probability and expectation under $H_\sigma^0$ and $H_\sigma'$, respectively.

Fix any $\gamma \in (\frac{2}{3}, 1)$, and let $\lambda$ be some constant such that

$$\mu_2 < \lambda \leq 1, \text{ and } |\mathrm{KL}(\mu_1 || \lambda) - \mathrm{KL}(\mu_1 || \mu_2)| \leq \gamma \, \mathrm{KL}(\mu_1 || \mu_2). \qquad (51)$$

Define event $\mathcal{N} := \left\{ N_{a_1} < \frac{1-\gamma}{\mathrm{KL}(\mu_1 || \lambda)} \ln T \right\}$.

Denote $Z_{a,i} \in \{0, 1\}$ as the $i$-th realization of playing arm $a$, and $\rho(x; \mu) = \mu^x (1-\mu)^{1-x}$ for $x \in \{0, 1\}$ and $\mu \in [0, 1]$ as the probability for Bernoulli random variables. Then we can define function

$$L(\vec{N}) := \ln \left( \prod_{i=1}^{N_{a_1}} \frac{\rho(Z_{a_1,i}; \mu_1)}{\rho(Z_{a_1,i}; \lambda)} \cdots \prod_{i=1}^{N_{a_{d-1}}} \frac{\rho(Z_{a_{d-1},i}; \mu_1)}{\rho(Z_{a_{d-1},i}; \lambda)} \cdot \prod_{i=1}^{N_{a_d}} \frac{\rho(Z_{a_d,i}; \mu_1)}{\rho(Z_{a_d,i}; \mu_3)} \right). \qquad (52)$$

Fix any $\eta \in (0, 3\gamma - 2)$, and define event $\mathcal{L} := \left\{ L(\vec{N}) \leq (1-\eta) \ln T \right\}$. In the following, we will show that

$$\mathbb{P}[\mathcal{N}] = \mathbb{P}\left[ N(a_1) < \frac{1-\gamma}{\mathrm{KL}(\mu_1 || \lambda)} \ln T \right] \to 0, \text{ as } T \to \infty, \qquad (53)$$

by proving that both $\mathbb{P}[\mathcal{N} \cap \mathcal{L}]$ and $\mathbb{P}[\mathcal{N} \cap \overline{\mathcal{L}}]$ tend to 0.

**Step 1: for** $\mathbb{P}\left[\mathcal{N} \cap \mathcal{L}\right]$**.** For $H'_\sigma$ ($\sigma$ is the greedy sequence), the sequence we play is the greedy sequence only if $a_1, \cdots, a_d$ are chosen simultaneously, thus the total number of playing $\sigma$ is $N(a_d)$. Furthermore, for any consistent algorithm, for any $\eta \in (0, \gamma)$, we have $\mathbb{E}'\left[T - N(a_d)\right] = o(T^\eta)$ under $H'_\sigma$. Since in our instance $N(a_1) \geq N(a_d)$ holds, thus $\mathbb{E}'\left[T - N(a_1)\right] \leq \mathbb{E}'\left[T - N(a_d)\right] = o(T^\eta)$.

$$
\begin{aligned}
\mathbb{P}'\left[\mathcal{N}\right] &= \mathbb{P}'\left[N(a_1) < \frac{1-\gamma}{\mathrm{KL}\left(\mu_1 || \lambda\right)} \ln T\right] \\
&= \mathbb{P}'\left[T - N(a_1) > T - \frac{1-\gamma}{\mathrm{KL}\left(\mu_1 || \lambda\right)} \ln T\right] \\
&\leq \frac{\mathbb{E}'\left[T - N(a_1)\right]}{T - \frac{1-\gamma}{\mathrm{KL}(\mu_1 || \lambda)} \ln T} \qquad\qquad\qquad \{\text{Markov's inequality}\} \\
&= o(T^{\eta-1}).
\end{aligned}
\tag{54}
$$

Let $\rho_x$ be the probability measure for $H^0_\sigma$, then through the change of probability measure, we can get that

$$
\begin{aligned}
\mathbb{P}'\left[\mathcal{N} \cap \mathcal{L}\right] &= \int_{x \in \mathcal{N} \cap \mathcal{L}} \prod_{i=1}^{N(a_1)} \frac{\rho(Z_{a_1,i}; \lambda)}{\rho(Z_{a_1,i}; \mu_1)} \cdots \prod_{i=1}^{N(a_{d-1})} \frac{\rho(Z_{a_{d-1},i}; \lambda)}{\rho(Z_{a_{d-1},i}; \mu_1)} \cdot \prod_{i=1}^{N(a_d)} \frac{\rho(Z_{a_d,i}; \mu_2 + \frac{1}{4})}{\rho(Z_{a_d,i}; \mu_1)} \, \mathrm{d}\rho_x \\
&= \int_{x \in \mathcal{N} \cap \mathcal{L}} \exp(-L(\vec{N})) \, \mathrm{d}\rho_x \tag{55} \\
&\geq T^{\eta-1} \int_{x \in \mathcal{N} \cap \mathcal{L}} \mathrm{d}\rho_x = T^{\eta-1} \, \mathbb{P}[\mathcal{N} \cap \mathcal{L}], \tag{56}
\end{aligned}
$$

where (55) is from (52), and (56) is because $\mathcal{L}$ holds. Thus, we can imply from (56) and (54) that

$$
\mathbb{P}[\mathcal{N} \cap \mathcal{L}] \leq T^{1-\eta} \cdot \mathbb{P}'\left[\mathcal{N} \cap \mathcal{L}\right] \leq T^{1-\eta} \cdot \mathbb{P}'\left[\mathcal{N}\right] = o(1).
\tag{57}
$$

**Step 2: for** $\mathbb{P}\left[\mathcal{N} \cap \overline{\mathcal{L}}\right]$**.** From Equation (52), we know that

$$
L(\vec{N}) = \sum_{i=1}^{N(a_1)} \ln\left(\frac{\rho(Z_{a_1,i}; \mu_1)}{\rho(Z_{a_1,i}; \lambda)}\right) + \cdots + \sum_{i=1}^{N(a_{d-1})} \ln\left(\frac{\rho(Z_{a_{d-1},i}; \mu_1)}{\rho(Z_{a_{d-1},i}; \lambda)}\right) + \sum_{i=1}^{N(a_d)} \ln\left(\frac{\rho(Z_{a_d,i}; \mu_2)}{\rho(Z_{a_d,i}; \mu_3)}\right)
\tag{58}
$$

$$
\to (N(a_1) + N(a_2) + \cdots + N(a_{d-1})) \, \mathrm{KL}(\mu_1 || \lambda) + N(a_d) \cdot \mathrm{KL}(\mu_1 || \mu_3) =: \overline{L}(\vec{N}),
\tag{59}
$$

as $N(a_1), \cdots, N(a_d)$ tend to be sufficiently large. Due to condition (48) and the definition of $\mathcal{N}$, we have

$$
\begin{aligned}
\overline{L}(\vec{N}) &\leq N(1) \left(1 + \frac{1}{W} + \cdots + \frac{1}{W^{d-2}}\right) \mathrm{KL}(\mu_1 || \lambda) + N(1) \frac{1}{W^{d-1}} \cdot \mathrm{KL}(\mu_1 || \mu_3) \\
&\leq N(1) \frac{1}{1 - \frac{1}{W}} \mathrm{KL}(\mu_1 || \lambda) + N(1) \frac{1}{W^{d-1}} \cdot \mathrm{KL}(\mu_1 || \mu_3) \\
&\leq \left(\frac{1}{1 - \frac{1}{W}} + \frac{1}{W^{d-1}} \frac{\mathrm{KL}(\mu_1 || \mu_3)}{\mathrm{KL}(\mu_1 || \lambda)}\right) (1 - \gamma) \ln T.
\end{aligned}
\tag{60}
$$

Since $2(x - y)^2 \leq \mathrm{KL}(x || y) \leq \frac{(x-y)^2}{y(1-y)}$ for $x, y \in [0, 1)$, $\frac{1}{4} < \mu_1 < \mu_2 < \mu_3 = \frac{3}{4}$, and $\mu_2 < \lambda \leq 1$, it can be implied that

$$
\frac{\mathrm{KL}(\mu_1 || \mu_3)}{\mathrm{KL}(\mu_1 || \lambda)} \leq \frac{(\mu_3 - \mu_1)^2}{\mu_3(1 - \mu_3) \cdot 2 \cdot (\lambda - \mu_1)^2} \leq \frac{3}{2(\mu_2 - \mu_1)^2}.
\tag{61}
$$

Assume that $W \geq 2$ and $W^{d-1} \geq \frac{2}{3(\mu_2 - \mu_1)^2}$, then from (60) and (61) we have $\overline{L}(\vec{N}) \leq 3(1 - \gamma) \ln T$.

Now fix any $\eta \in (0, 3\gamma - 2)$, and it is easy to know that $\overline{L}(\vec{N}) \leq 3(1-\gamma)\ln T < (1-\eta)\ln T$. Recall that $L(\vec{N}) \to \overline{L}(\vec{N})$ from (59) and $\mathcal{N} = \{N(a_1) < \frac{1-\gamma}{\mathrm{KL}(\mu_1||\lambda)}\ln T\} \subsetneq \bigcup_{j=0}^{\frac{1-\gamma}{\mathrm{KL}(\mu_1||\lambda)}\ln T} \{N(a_1) = j\}$. Then, by the strong law of large numbers, we can get

$$\mathbb{P}\left[\mathcal{N} \cap \overline{\mathcal{L}}\right] = \mathbb{P}\left[N(a_1) < \frac{1-\gamma}{\mathrm{KL}(\mu_1||\lambda)}\ln T \ \wedge \ L(\vec{N}) > (1-\eta)\ln T\right] \to 0, \text{ as } T \to \infty. \quad (62)$$

**Step 3: combine two parts.** Because $d \geq \log_W\left(\frac{2W}{3(\mu_2-\mu_1)^2}\right)$, we have that $W^{d-1} \geq \frac{3}{2(\mu_2-\mu_1)^2}$. For any $\gamma \in (\frac{2}{3}, 1)$, it can be derived from (57) and (62) that $\lim_{T\to\infty}\mathbb{P}\left[\mathcal{N}\right] = \lim_{T\to\infty}\mathbb{P}\left[N_T(a_1) < \frac{1-\gamma}{\mathrm{KL}(\mu_1||\lambda)}\ln T\right] = 0$. Then, from (51), we can get

$$\lim_{T\to\infty}\mathbb{P}\left[N(a_1) < \frac{1-\gamma}{(1+\gamma)\,\mathrm{KL}(\mu_1||\mu_2)}\ln T\right] = 0. \quad (63)$$

Therefore,

$$\lim_{t\to+\infty}\frac{\mathbb{E}\left[N_t(a_1)\right]}{\ln t} \geq \frac{0.39}{\mathrm{KL}(\mu_1||\mu_2)}. \quad (64)$$

$\square$

We will show that the greedy regret can be as large as $\Omega(\frac{mW\log T}{\Delta^2})$, when (1) the unit gap ($\mu_2 - \mu_1$) is not exponentially small or (2) $m$ is large enough, which will be specified quantitatively in Theorem E.2.

**Theorem E.2.** *For instance $\mathcal{P}$ of the prize-collecting problem, for any $\mu_1, \mu_2, \mu_3$ satisfying that $\frac{1}{4} < \mu_1 < \mu_2 = \frac{1}{2} < \mu_3 = \frac{3}{4}$ and $W \geq 2$, assume that there exists some constant $\xi \in (0,1)$ such that $\xi m \geq \log_W\left(\frac{2W}{3(\mu_2-\mu_1)^2}\right)$ (i.e., $(\mu_2-\mu_1)^2 \geq \frac{2}{3W^{\xi m-1}}$). Denote $\Delta = \mu_2 - \mu_1 > 0$. For any consistent algorithm, as time $T$ tends to $+\infty$, the greedy regret satisfies that*

$$R^{\mathsf{G}}(T) \geq \frac{0.39 \cdot (1-\xi)m(W-1)\Delta_{\min}}{\mathrm{KL}(\mu_1||\mu_2)}\ln T, \quad (65)$$

*where $\mathrm{KL}(p||q) = p\ln\frac{p}{q} + (1-p)\ln\frac{1-p}{1-q}$ is Kullback-Leibler divergence for Bernoulli random variables, and $\Delta_{\min} = \Delta + \frac{1}{4}$.*

*Proof.* From the assumption, we know that $\xi m \geq \log_W\left(\frac{2W}{3(\mu_2-\mu_1)^2}\right)$. Fix any $\ell = 1, 2, \cdots, (1-\xi)m$, then it is easy to see that the deviated $d = m - \ell$ arms (denoted as $a_1, a_2, \cdots, a_d$) satisfying $d \geq \log_W\left(\frac{2W}{3(\mu_2-\mu_1)^2}\right)$.

Denote the true greedy sequence as $\sigma^{\mathsf{G}} = \langle G_0, G_1, \cdots, G_m \rangle$. For any sequence $\sigma$ coinciding with $\sigma^{\mathsf{G}}$ for the prefix $\langle G_0, G_1, \cdots, G_\ell \rangle$, for phase $\ell+1$, we can choose $W-1$ candidates from $\mathcal{N}(G_\ell) \setminus \{g_{G_\ell}^*\}$. According to Lemma E.1, for each candidate, the consistent algorithm needs to play arm $a_1$ for at least $\frac{0.39}{\mathrm{KL}(\mu_1||\mu_2)}\ln T$ times. Since the penalty is at least $\Delta_{\min} = \Delta + \frac{1}{4}$, therefore we can conclude that

$$R^{\mathsf{G}}(T) \geq \sum_{\ell=1}^{(1-\xi)m}(W-1)\cdot\frac{0.39 \cdot \Delta_{\min}}{\mathrm{KL}(\mu_1||\mu_2)}\ln T = \frac{0.39 \cdot (1-\xi)m(W-1)\Delta_{\min}}{\mathrm{KL}(\mu_1||\mu_2)}\ln T. \quad (66)$$

$\square$

From Theorem E.2 and $\mathrm{KL}(\mu_1||\mu_2) \leq \frac{(\mu_2-\mu_1)^2}{\mu_2(1-\mu_2)} = 4\Delta^2$, it is easy to check that the greedy regret is $\Omega\left(\frac{mW\Delta_{\min}}{\Delta^2}\log T\right)$. Since $\Delta_{\max} = m\Delta + \frac{1}{4}$ and $\Delta_{\min} = \Delta + \frac{1}{4}$ in the setting of Theorem E.2, suppose we further assume that $\Delta = \mu_2 - \mu_1 = o(\frac{1}{m})$ (e.g., $\Delta = \frac{0.01}{m^{1.1}}$), then it indicates that $\Delta_{\max} = \Delta_{\min} + o(1)$. Thus, we can get that the lower bound $R^{\mathsf{G}}(T) = \Omega\left(\frac{mW\Delta_{\max}}{\Delta^2}\log T\right)$, which matches with the upper bound up to a constant factor.

### E.2 Lower bound with an exponential term

We further remark that a similar problem instance as in Appendix E.1 can be used to show a regret lower bound with an exponential term: $\Omega\left(\frac{W^m}{(\mu_3-\mu_2)}\log T\right)$.

The intuition is that all the first $m-1$ layers have essentially no difference in reward for any decision sequence, and all reward is on the last layer, i.e., $\mu_3 - \mu_2 \gg \mu_2 - \mu_1 > 0$ and $\mu_2 - \mu_1 = O(\frac{1}{W^m})$, which makes it similar to the classical multi-armed bandit with $W^m$ arms. Therefore, the lower bound is in the order of $\Omega\left(\frac{W^m}{(\mu_3-\mu_2)}\log T\right)$. This means we cannot have $O\left(\frac{\text{poly}(W,m)}{(\mu_3-\mu_2)}\log T\right)$ regret.

## F  Extensions

### F.1  A simple extension of OG-UCB for linear bandits with a matroid constraint to achieve $O(n/\Delta \log T)$ regret and $O(n)$ space

If we know that the problem instance has linear reward function, i.e., $f_t(S) = \sum_{e \in S} f_t(e)$, and the accessible set system $(E, \mathcal{F})$ is restricted to a matroid, we can easily extend OG-UCB and make it behave essentially the same as Algorithm OMM in [23].

The key is to merge those equivalent arms. More formally, we call two arms $a = e|S$ and $a' = e|S'$ *equivalent* if the marginal rewards of both arms follow the same distribution. For these equivalent arms, we merge estimator $\hat{X}(a)$ and $\hat{X}(a')$ (and the counters $N(a)$ and $N(a')$), and it can be achieved simply by using the same memory for $a$ and $a'$. This applies to the setting of linear reward function [23]: the marginal value of choosing arm $e|S$ or $e|S'$ only depends on element $e$ and is irrelevant to $S$ or $S'$, and thus we merge all such arms $e|S$ and $e|S'$, such that observations of one arm refine the estimation of its equivalent arms. In this case, it can be easily verified that OG-UCB and OMM behave essentially the same (except on some minor manipulation of time counter $t'$ in Line 4 of Algorithm 1).

In this case, we remark that: (1) this extension utilizes the property of linear bandits and matroid constraints; (2) the greedy algorithm will find the optimal solution, and henceforth the greedy regret is also equivalent to the expected cumulative regret (i.e., $R^1(T)$). Therefore, the analysis of OMM in [23] applies to OG-UCB: OG-UCB has regret bound $O(\frac{n}{\Delta}\log T)$ and the memory cost is $O(n)$, where $\Delta$ is the minimum unit gap.

### F.2  Algorithms for Knapsack Constraints

In this section, we consider the knapsack constraint, which is a special case of our accessible set system $(E, \mathcal{F})$.

For each element $e \in E$, a cost function $\lambda(e) \in \mathbb{R}_{\geq 0}$ is given in advance. Without loss of generality, we assume $\lambda(a) = \lambda(e) \in [1, \infty)$ for each arm $a = e|S \in \mathcal{A}$. Given a budget $B$ for the knapsack problem, the collection of feasible sets are $\mathcal{F} = \{S \subseteq E : \sum_{e \in S} \lambda(e) \leq B\}$. As is described Example A.4, such $(E, \mathcal{F})$ is an accessible set system.

Notice that, for each phase, suppose the offline greedy still maximizes the marginal reward $\overline{f}(a)$ from a set of accessible arms as before, then it is only the special case of the original model described in the main text, therefore the algorithms and results apply without any change.

However, for many problems with knapsack constraints, a natural way of the greedy algorithm is to maximize the marginal reward per unit cost every phase i.e., $\frac{\overline{f}(a)}{\lambda(a)}$ is maximized (instead of maximizing the marginal reward of a set accessible arms). We show that we can slightly modify our previous algorithm to accommodate this setting.

In particular, our algorithm can be slightly modified as follows: in the offline and online problems, $\arg\max$ and $\text{MaxOracle}$ return an arm $a$ such that $\frac{\hat{X}(a)}{\lambda(a)}$ is maximized (replace $\hat{X}(a)$ in the objective

of $\arg\max$ and $\mathrm{MaxOracle}$ with $\frac{\hat{X}(a)}{\lambda(a)}$). Then, we can rewrite the *greedy element*:

$$\forall S \in \mathcal{F}, \quad g_S^* := \arg\max_{e \in \mathcal{N}(S)} \frac{\overline{f}(e|S)}{\lambda(e|S)}, \tag{67}$$

together with the *unit gap*:

$$\forall S \in \mathcal{F}, \forall e \in \mathcal{N}(S), \quad \Delta(e|S) := \begin{cases} \frac{\overline{f}(g_S^*|S)}{\lambda(g_S^*|S)} - \frac{\overline{f}(e|S)}{\lambda(e|S)}, & e \neq g_S^* \\ \frac{\overline{f}(e|S)}{\lambda(e|S)} - \max_{e' \in \mathcal{N}_-(S)} \frac{\overline{f}(e'|S)}{\lambda(e'|S)}, & e = g_S^*. \end{cases} \tag{68}$$

The algorithms OG-UCB and OG-LUCB with such modification still work. The rest of the analysis is the same. Therefore, Theorems 3.1, 4.1, 4.2 and 4.3 still apply after rewriting the above definitions.

# G    Applications

## G.1    Top-$m$ Selection with a Consistent Function

In this subsection, we will demonstrate the application of our framework to the problem of Top-$m$ Selection with a consistent reward function.

Generally speaking, the reward of this problem may be non-linear, and the greedy algorithm can output an optimal solution (with approximation ratio $\alpha = 1$).

**The decision class.**    In the top-$m$ selection problem, the set of feasible sets are all subsets of cardinality at most $m$, hence corresponding to an accessible set system $(E, \mathcal{F})$, where $E = \{1, 2, \cdots, n\}$, and $\mathcal{F} = \{S \subseteq E : |S| \leq m\}$.

Notice that, the arm space is $\mathcal{A} = \{e|S : \forall S, \forall e \in \mathcal{N}(S), |S| \leq m - 1\}$, containing $\prod_{i=0}^{m-1}(n - i)$ arms in total.

**Definition G.1** (Consistent function [5]). A function $\psi : 2^E \to \mathbb{R}$ is *consistent* if for any $T \subset T' \subset E$, and element $x, y \in E \setminus T'$, we have $\psi(T \cup \{x\}) \geq \psi(T \cup \{y\}) \implies \psi(T' \cup \{x\}) \geq \psi(T' \cup \{y\})$.

**Reward function.**    Let $\Omega \subseteq [0,1]^E$ be the probability space. For each time $t = 1, 2, \ldots$, the environment draws an i.i.d. samples $\omega_t$ from $\Omega$. $\omega_t$ can be thought as a vector and $\omega_t(e) \in [0,1]$ for each dimension $e \in E$. For any $\omega_t \in \Omega$, the reward function is a bounded and non-decreasing function $f_t(S) := f(S, \omega_t) \in [L, L + m]$ (fix constant $L \geq 0$) as required in Section 2, and define $\overline{f}(S) = \mathbb{E}[f_t(S)]$ where the expectation is taken over $\omega_t$. We assume that $\overline{f}(\cdot)$ is a *consistent function*.

**Offline and online settings.**    In the offline setting, $\overline{f}(\cdot)$ is provided as a value oracle and we would like to return the subset with the maximum $\overline{f}$ value. In the online setting, the player first plays a decision sequence $\sigma = \langle S_0, S_1, \cdots, S_k \rangle$ with $S_k = S_{k-1} \cup \{s_k\}$ for time $t$, then observes the semi-bandit feedbacks $\{f_t(S_i)\}_{i=0,\ldots,k}$, and gains a reward of $f_t(S_k)$.

**Example G.1.** The following functions belong to the consistent function family:

- $\overline{f}(S) = c \cdot e^{\sum_{e \in S} \mathbb{E}[\omega_t(e)]}$ ($c$ is a small constant for normalization);

- $\overline{f}(S) = 1 - \prod_{e \in S}(1 - \mathbb{E}[\omega_t(e)])$;

- $\overline{f}(S) = \frac{1}{2}(|S| + \min_{e \in S} \mathbb{E}[\omega_t(e)])$, with $f_t(\emptyset) = 0$. It is a variant of bottleneck functions, and the use $|S|$ and $\frac{1}{2}$ is for normalization;

- $\overline{f}(S) = \sum_{e \in S} \mathbb{E}[\omega_t(e)]$ (linear function).

Denote the greedy sequence for the top-$m$ selection problem as $\sigma^{\mathsf{G}} = \langle G_0, G_1, \cdots, G_m \rangle$ ($m$ is the length). from Theorem 2 of [5]:

**Lemma G.1.** *For consistent function $\overline{f}(\cdot)$, $G_m$ is an optimal solution, i.e., $\overline{f}(G_m) = \max_{\forall S \in \mathcal{F}} \overline{f}(S)$ is satisfied.*

We can use our algorithm OG-UCB to solve the online problem by playing the game during the time horizon $T$. Since the offline greedy solution is also the optimal one (approximation ratio $\alpha = 1$), from Theorem 3.1 and Proposition B.1, we have the following corollary:

**Corollary G.2.** *For any time $T$, for the top-$m$ selection problem with a consistent function, our algorithm OG-UCB can achieve a regret*

$$R(T) \leq R^{\mathsf{G}}(T) \leq \sum_{a \in \Gamma_-(\sigma^{\mathsf{G}})} \left( \frac{6\Delta^*(a) \cdot \ln T}{\Delta(a)^2} + \left( \frac{\pi^2}{3} + 1 \right) \Delta^*(a) \right), \tag{69}$$

*where $\sigma^{\mathsf{G}}$ is the greedy sequence of length $m$, and $R(T)$ is the expected regret with respect to the optimal solution.*

Notice that $\Gamma_-(\sigma^{\mathsf{G}})$ has $\sum_{i=1}^{m}(n-i) = (n-1-\frac{m}{2})m$ arms in the above corollary for the top-$m$ selection.

We claim that other decision classes belonging to a matroid embedding [5] also work through the analysis. (The key is to derive the optimal solution guarantee analogue to Lemma G.1.) To the best of our knowledge, the bound of $O(\log T)$ regret for the online version of the top-$m$ selection problem (or the wider decision classes) with a general consistent function is new.

### G.2 Stochastic Online Submodular Maximization

Our algorithms and results can be applied to online submodular maximization. For any time $t$ and for any $S \in \mathcal{F}$, assume the reward function $f_t(S) \in [L, L+m]$ $(L \geq 0)$ is a non-negative monotone submodular function. Denote $\overline{f}(S) := \mathbb{E}[f_t(S)]$, and it is provided for the offline problem. It is obvious that $\overline{f}(\cdot)$ is also non-negative monotone submodular due to the linearity of the expectation.

Let the accessible set system $(E, \mathcal{F})$ be a uniform matroid, in which the maximal feasible set $S \in \mathcal{F}$ is of size $m$. We adapt the result for offline submodular maximization in [21, 28], with a slight generalization for the greedy (or $\epsilon$-quasi greedy) policy (The proof is almost identical to that in [21], and is omitted here).

**Lemma G.3.** *Assume that the non-negative monotone submodular function $\overline{f}(\cdot)$ is provided as a value oracle. For any $\epsilon \geq 0$, suppose a decision sequence $\sigma = \langle S_0, S_1, \cdots, S_k \rangle \in \mathcal{F}^{k+1}$ satisfies $S_0 = \emptyset$, $S_i = S_{i-1} \cup \{s_i\}$ for every $i$, and $\overline{f}(s_i | S_{i-1}) \geq \overline{f}(g_{S_{i-1}}^* | S_{i-1}) - \epsilon$ for every $i$. Then, for every positive integer $\ell$ and $m$,*

$$\overline{f}(S_\ell) \geq \left( 1 - \left( 1 - \frac{1}{m} \right)^k \right) \left( \overline{f}(S^*) - m\epsilon \right), \tag{70}$$

*where $k$ is the length of the decision sequence and $S^* = \arg\max_{S \in \mathcal{F}} \overline{f}(S)$.*

Noting that $1 - x \leq e^{-x}$ for all $x \in \mathbb{R}$, we can immediately derive the following two properties from the above lemma.

**Lemma G.4.** *(1) Let $\sigma^{\mathsf{G}} = \langle S_0, S_1, \cdots, S_m \rangle \in \mathcal{F}^{m+1}$ be the greedy sequence of size $m$. Then,*

$$\overline{f}(S_m) \geq \left( 1 - e^{-1} \right) \overline{f}(S^*).$$

*Thus, $S_m$ in $\sigma^{\mathsf{G}}$ is an $\alpha$-approximation solution for $\alpha = 1 - e^{-1}$.*

*(2) Given any small $\epsilon$ such that $0 \leq \epsilon \leq \frac{L}{m}$, let $\sigma^{\mathsf{Q}} = \langle S_0, S_1, \cdots, S_m \rangle \in \mathcal{F}^{m+1}$ be the minimum of $\epsilon$-quasi greedy sequence. Then,*

$$\overline{f}(S_m) \geq \left( 1 - e^{-1} - \frac{m\epsilon}{L} \right) \overline{f}(S^*).$$

*Thus, $S_m$ in $\sigma^{\mathsf{Q}}$ is an $\alpha$-approximation solution for $\alpha = 1 - e^{-1} - \frac{m\epsilon}{L}$.*

Now we can use Algorithm OG-UCB and OG-LUCB to solve the online version of the submodular maximization problem. From Propositions B.1 and B.2, we know that $R^\alpha(T) \leq R^G(T)$ and $R^\alpha(T) \leq R^Q(T)$ for the respective $\alpha$-approximation solutions. Then, from Theorem 3.1 and Theorem D.7, and we have the following $\alpha$-approximation regret:

**Corollary G.5.** *For any time horizon $T$, for the online submodular maximization problem, our algorithm OG-UCB can achieve the following $\alpha$-approximation regret:*

$$R^\alpha(T) \leq R^G(T) \leq \sum_{a \in \Gamma_-(\sigma^G)} \left( \frac{6\Delta^*(a) \cdot \ln T}{\Delta(a)^2} + \left( \frac{\pi^2}{3} + 1 \right) \Delta^*(a) \right), \tag{71}$$

*where $\sigma^G$ is the greedy sequence, and $\alpha = 1 - e^{-1}$ is the approximation ratio of the offline problem.*

**Corollary G.6.** *For the online submodular maximization problem, suppose we run Algorithm OG-LUCB-R with some small $\epsilon$ such that $0 \leq \epsilon \leq \frac{c}{m}$, and let $\alpha = 1 - e^{-1} - \frac{m\epsilon}{L}$ be the approximation ratio of the offline problem.. Use $\phi_i$, $\tau_i$, $\sigma^{(\ell)}$, $K$, $R^{Q,\sigma^{(\ell)}}(\cdot)$, $c_\ell$ and $b_\ell$ defined in Theorem D.7. Our algorithm OG-LUCB-R can achieve an $\alpha$-approximation regret as follows:*

$$R^\alpha(T) \leq R^Q(T) \leq \sum_{\ell=1}^{K} R^{Q,\sigma^{(\ell)}}(\phi_\ell). \tag{72}$$

*Furthermore, we can get*

$$R^\alpha(T) \leq R^Q(T) \leq 4 \cdot \left( \max_{\ell=1,\cdots,K} c_\ell \right) \cdot \ln(T) + \left( \max_{\ell=1,\cdots,K} b_\ell \right) \cdot \log_2 \left( 2\ln T \right). \tag{73}$$

Besides, the exploration time for Algorithm OG-LUCB$_{\epsilon,\delta}$ for any input of $(\epsilon, \delta)$ is the same as Theorem 4.1; and letting $\delta = \frac{1}{T}$ implies a corollary for the $\alpha$-approximation regret $R^\alpha$ ($R^\alpha \leq R^Q$ and $\alpha = 1 - e^{-1}$), corresponding to Theorem D.5, both of which will not be restated here.

The above results for stochastic online greedy, ensuring the $O(\log T)$ bounds, are complementary to the $O(\sqrt{T})$ $\alpha$-approximation regret bound presented in [31], which focuses on the online submodular maximization in the adversarial setting.

### G.3 Online Influence Maximization and Probabilistic Set Cover in a Unified Model

From the above Appendix G.2, we have shown that our algorithms work without assuming the detailed form of the reward function. In light of this, our model can be applied to unify the online *Influence Maximization (IM)* and *Probabilistic Set Cover (PMC)* problems [19] across different models. (PMC can be viewed as two-layers influence maximization on the independent cascade model.)

Influence maximization problem [19] has been widely studied in the viral marketing. In general, given social graph $\mathcal{G} = (\mathcal{V}, \mathcal{E})$ representing nodes and edges respectively, the goal is to choose a $m$-nodes set $S \subseteq \mathcal{V}$ as *seeds*, so that starting from the seed nodes, at many nodes as possible are influenced (or covered) via the word-of-mouth effect. Many models, such as the *independent cascade (IC)* model, the *linear threshold (LT)* model [19] and the *continuous-time independent cascade (CIC)* model [30], are proposed to capture different natures of users' behaviors. The common part of those models (IC, LT, CIC, etc.) is that they assume that the coverage function $f : 2^\mathcal{V} \times \Omega \to \mathbb{R}$ (and we use $f_t(S) = f_t(S, \omega_t)$ for convenience, where $\omega_t$ is the randomness of the environment at time $t$), and focus on maximizing the expected coverage $\overline{f}(S) := \mathbb{E}[f_t(S)]$ called *influence spread*, which is a non-negative monotone submodular function. Note that due to the #P-hardness of computing $\overline{f}(\cdot)$ [10] in order to find the optimal solution, Monte Carlo simulations on $f_t(\cdot)$ are usually carried out to estimate $\overline{f}(\cdot)$. The difference of models lies in the assumption of the environmental randomness $\Omega$: in IC model, it comes from independent Bernoulli random variables assigned on edges; CIC model parametrizes edges with time-variant distributions, and a constant total cutting time; and in LT model, the randomness are from the thresholds of nodes. The empirical study in [13] shows that the model misspecification may lead to one optimal solution for one model yielding a bad performance in another.

In the online version of influence maximization, we need to choose the seeds set $S$ over time to maximize the overall performance against the optimal $S^*$ in hindsight. [11] shows how to model the online influence maximization for IC model as a combinatorial multi-arm bandit problem. If the IC model is assumed, the randomness of edges can be modelled as base arms and the reward is a function represented by the expected values of those arms. However, in our framework, we do not need the knowledge of the detailed diffusion model or the exact form of the reward function. For different influence maximization models, we only access the value of $f_t(\cdot)$ without assuming the exact form $f_t(\cdot)$ or $\overline{f}(\cdot)$. The application of our framework, utilizing value oracle $f_t(\cdot)$, can unify the above models of online influence maximization as well as avoid the risk of model misspecification. As is illustrated in Appendix G.2, our algorithms still apply here and the same results hold.

## H   Empirical Evaluation for the Lower Bound on the Prize-Collecting Problem

In this section, we carry out an empirical evaluation for the prize-collecting problem to validate Algorithms OG-UCB. We use the problem instance $\mathcal{P}$ as described in Appendix E.

**Setup.**   We evaluate the performance over the following combinations: (1) Select one out of $W \in \{10, 20, 30\}$ elements, for each of $m \in \{4, 6, 8\}$ bandits; (2) The minimum unit gap $\Delta$ is chosen from $\{0.2, 0.1\}$ (that is, $\mu_1 \in \{0.3, 0.4\}$ and $\mu_2 = 0.5$ with $\Delta = \mu_2 - \mu_1$), and $\mu_3 = 0.75$.

For each case, we set the same total horizon $T = 10^6$. The lower bound (LB) is estimated by the right-hand side of (65) in Theorem E.2.

The experiment is repeated for 20 times, and we calculate the average of the regret together with their standard deviation.

**Analysis.**   Table 2 illustrates the result of Algorithms OG-UCB. The result contains three columns of values: The left column is regrets of OG-UCB with their standard deviation in absolute values $(\times 10^4)$, the middle column is the lower bound estimated in absolute values $(\times 10^4)$, and the right column (surrounded by parentheses) is the ratio between regrets in practice and LB.

First, observing the ratio between regrets and LB, we know that it is about 20 times. The ratio does not increase with $W$ or $m$. Therefore, it indicates that our upper bound matches with the lower bound, and they are tight up to a constant factor.

Second, comparing the regret of algorithms with the change of $W$ or $m$, we see that the regret increase linearly with $W$ or $m$. It matches with the regret $\Theta(\frac{Wm\Delta_{\max}}{\Delta^2} \log T)$ derived from our theoretical analysis. When $\Delta$ is changed from $0.2$ to $0.1$, the regret is also about $4$ times of the original. (It is less than $4$ times because $\Delta_{\max}$ shrinks meanwhile.)

Table 2: Experiment result of Algorithms OG-UCB for the prize-collecting problem ($\mathcal{P}$).

| $W$ | $m$ | $\Delta$ | OG-UCB ($\times 10^4$) | LB ($\times 10^4$) | (OG-UCB/LB) |
|---|---|---|---|---|---|
| 10 | 4 | 0.20 | $1.17 \pm 0.06$ | 0.047 | (24.8) |
| 10 | 4 | 0.10 | $2.80 \pm 0.12$ | 0.099 | (28.3) |
| 10 | 6 | 0.20 | $2.40 \pm 0.07$ | 0.100 | (23.9) |
| 10 | 6 | 0.10 | $5.56 \pm 0.19$ | 0.268 | (20.8) |
| 10 | 8 | 0.20 | $3.88 \pm 0.14$ | 0.153 | (25.3) |
| 10 | 8 | 0.10 | $9.00 \pm 0.26$ | 0.436 | (20.6) |
| 20 | 4 | 0.20 | $2.45 \pm 0.05$ | 0.115 | (21.3) |
| 20 | 4 | 0.10 | $6.01 \pm 0.16$ | 0.284 | (21.1) |
| 20 | 6 | 0.20 | $4.99 \pm 0.12$ | 0.227 | (21.9) |
| 20 | 6 | 0.10 | $11.54 \pm 0.32$ | 0.640 | (18.0) |
| 20 | 8 | 0.20 | $8.24 \pm 0.17$ | 0.339 | (24.3) |
| 20 | 8 | 0.10 | $18.55 \pm 0.34$ | 0.996 | (18.6) |
| 30 | 4 | 0.20 | $3.78 \pm 0.08$ | 0.186 | (20.4) |
| 30 | 4 | 0.10 | $9.04 \pm 0.25$ | 0.479 | (18.9) |
| 30 | 6 | 0.20 | $7.59 \pm 0.10$ | 0.357 | (21.3) |
| 30 | 6 | 0.10 | $17.55 \pm 0.40$ | 1.023 | (17.2) |
| 30 | 8 | 0.20 | $12.61 \pm 0.17$ | 0.528 | (23.9) |
| 30 | 8 | 0.10 | $28.23 \pm 0.38$ | 1.566 | (18.0) |