[Reviews · NeurIPS 2015]

Submitted by Assigned_Reviewer_1

Strengths:

1) Technical and well executed. Upper bounds and a matching lower bound.

2) Pleasure to read.

3) A general class of problems which is of interest in large-scale learning.

Weaknesses:

1) Positioning with respect to the prior work is insufficient and needs to be improved.

2) Not particular novel and surprising, given the vast amount of work on related subjects in the past few years.

Major comments:

This paper cites most papers from the past few years on related subjects. However, it falls short in discussing details in which this work differs from the past work, such as the rates of regret bounds and the computational complexity of the proposed methods. This needs to be improved. This is my point of view. Matroids and polymatroids are generally regarded as the largest classes of problems where the greedy algorithm is guaranteed to be optimal. [23] and:

Kveton, Branislav, Wen, Zheng, Ashkan, Azin, and Valko, Michal. Learning to act greedily: Polymatroid semi-bandits. CoRR, abs/1405.7752, 2014.

which I further refer to as [33], proposed optimal learning algorithms for these problems. The main difference in this work is that the optimized function is not modular and that the feasible set is more general. In this setting, the greedy algorithm is not guaranteed to be optimal, or even to be a good approximation. This should be highlighted. Nevertheless, this work allows for solving new and interesting learning problems, such as learning to maximize a submodular function subject to a cardinality constraint in the semi-bandit setting.

In the notation of this paper, [23] and [33] prove O(n (1 / \Delta) \log n) upper bounds on the regret of their learning algorithms. Let's compare this to Theorem 3.1. Clearly W = O(n) and \Delta_{max} = O(m). Therefore, the upper bound in Theorem 3.1 is O(m^2 n (1 / \Delta^2) log n), a factor of m^2 (1 / \Delta) larger than the result of [23] on more specific problems. Perhaps this is not surprising, given how general the proposed framework is, but it should be mentioned that OG is less statistically efficient than OMM [23] on many practical problems, such as learning a minimum spanning tree. The authors should also discuss the space complexity of OG. As far as I can tell, it is linear in |F|, which is n \choose m when the feasible set is a uniform matroid of rank m. Therefore, OG is not space efficient on many practical problems. When the gaps are small, even lazy initialization (line 269) could lead to exploring all n \choose m arms. In comparison, the space complexity of OMM [23] on a uniform matroid of rank m is O(n).

Finally, note that [24] prove a O(m n (1 / \Delta') \log n) upper bound on the regret of CombUCB1 in stochastic combinatorial semi-bandits, where \Delta' is arguably larger than \Delta in this paper. Therefore, the upper bound of [24] is a factor of m (1 / \Delta) smaller than that in Theorem 3.1. This upper bound holds for any linear objective on any feasible set, and even for approximation oracles. Therefore, CombUCB1 is more statistically efficient than OG in this setting. The above also indicates that the extra factors in Theorem 3.1 may be due to the non-linearity of the optimized objective. This is worth mentioning.

Based on the above, the novelty of this work is mostly in learning suboptimal greedy policies for non-linear objectives. The proposed learning algorithms are not guaranteed to be space efficient. Therefore, the impact of this work on practice may be limited. To make it more significant, I suggest the authors include a good motivating example, which showcases the importance of learning good suboptimal greedy policies, which can be learned efficiently by either OG or LUCB. Experimental results on this problem would be nice and convince me that the class of problems in this paper is important and worth studying. The current motivating example is bad in the sense that this paper does not solve this particular problem.

My last major comment is that the lower bound in Section 5 is not very clear. In particular, I am not sure how the authors conclude that:

\Delta + 0.5 = \Delta_{max} + o(1).

It can be also argued that the case where \epsilon < \Delta is not very interesting.

Minor comments:

Line 128: It may be less ambiguous to write ordered sets as vectors and then abuse the set notation when needed. For instance, \sigma = (S_0, S_1, \dots, S_k) instead of \sigma = \{S_0, S_1, \dots, S_k\}.

Line 191: "online algorithm that are" should read as "online algorithms that are".

Line 192: "objective is to minimum" should read as "objective is to minimize".
Summary: This paper studies the problem of learning greedy solutions in stochastic combinatorial semi-bandits. The paper is well executed and should be accepted, although it is not particularly novel given the vast amount of work on related subjects in the past few years. Positioning with respect to the prior work is insufficient and needs to be improved.

Submitted by Assigned_Reviewer_2

In this paper, the authors have proposed two online greedy algorithms, OG-UCB and OG-LUCB, for a class of (stochastic) online learning problems. They have also derived regret bounds for these two algorithms under various assumptions (Theorem 3.1, 4.2, and 4.3), and claimed that a matching lower bound is achieved (Section 5). No experiment results are reported in this paper.

This paper is very interesting in general, however, I do not think it has met the (very high) standard of NIPS since the regret bounds in this paper are unsatisfactory. In particular, all the regret bounds are O(1/\Delta^2), where \Delta is some appropriate "gap". My understanding is that in almost all the stochastic bandit literature, the problem-dependent regret bounds are O(1/\Delta) rather than O(1/\Delta^2). Usually, the factor 1/\Delta^2 comes from some concentration inequality, and one \Delta is cancelled out when computing the regret bound. It will be quite surprising if the authors can show that the 1/\Delta^2 factor is intrinsic to this problem, and I believe that the authors can further polish their analysis to achieve O(1/\Delta) regret bounds for this problem. I would like to emphasize that showing an upper bound is tight in some problem instances does not guarantee that the upper bound is good in general cases. In the rebuttal, I hope the authors can provide EITHER (1) O(1/\Delta) regret bounds, OR (2) a detailed explanation why the 1/\Delta^2 factor is intrinsic to the general cases of this problem.

Some other comments:

1) In this paper an arm is defined as a set-item pair e|S (see Line 140). If we view S as a context, it seems that the considered problem is closely related to the contextual bandits. The authors should discuss the similarities/differences between this problem and the contextual bandits.

2) The authors said that "we are the first to propose the framework using the greedy regret...", this is not true. A recent ICML paper uses similar performance metrics:

http://jmlr.org/proceedings/papers/v37/wen15

3) The authors should rewrite Section 5 to improve its readability.

***************************************************************************************************************

I have read the authors' rebuttal. The authors have clarified the upper bound and the lower bound. I agree with the authors that (1) when restricted to linear reward function, the proposed algorithm can achieve

O(1 / \Delta \log T) regret; (2) the \Omega(1/\Delta^2 \log T) lower bound is intrinsic to some problems with nonlinear reward function.

My major concern is that given the existing literature, it is not very surprising that the authors can derive an O(1 / \Delta^2 \log T) upper bound. But showing that \Omega(1/\Delta^2 \log T) lower bound is intrinsic to some problems with nonlinear reward function seems to be a contribution to the ML community. Thus, I have changed my score to 6.
Summary: This paper is very interesting in general, though it is not very novel given the existing literature. I think it is marginally above the acceptance threshold.

Submitted by Assigned_Reviewer_3

The paper discusses the multi-armed bandit problem, in a stochastic setting. It focuses on the setup where the arms are in fact members of a family of subsets of some pre-defined set. Rather than competing with the subset with the highest corresponding expected reward, the algorithms compete with a subset obtained by knowing the rewards associated with the different subsets and applying a greedy algorithm that adds elements to the subsets one at a time in a greedy manner. This type of guarantee makes sense when something provable can said about the greedy algorithm, which is indeed a common scenario (for example if the reward is monotone-submodular). The authors provide a further extension that competes with a slightly weaker adversary that chooses its set in an approximately greedy fashion.

The authors provide a formal definition of the problem setup, and provide algorithms for both of the variants above, both achieving a problem dependent regret bound of log(T).

The paper is overall well written (with the exception of the presence of more than a few grammar errors such as a missing 'a', etc). Its proofs hold. They use machinery that is by now standard in bandit-related papers. The results in the paper are honestly and accurately stated. The problem setting and the ideas behind the proofs are clear and written in an accessible manner.

The definition of the feedback provided when selecting a set lacks motivation: It is assumed that the revealed data contains the rewards is for every subset in a sequence of sets starting from the empty set, ending in the chosen one. What is the motivation for this definition? Is there any scenario where this is the natural form of the output? I should mention that a different setting that requires no explanation is that when the feedback contains only the value associated with the chosen set. The techniques could probably be modified to handle this type of change at the cost of slightly worse second order terms.

The problem being dealt with is interesting and has proper motivation, though previous papers have dealt it with in the past, in one way or another. There are many examples in which the greedy approach is known either to provide proven guarantees compared to the optimal solution, in the setting of choosing a subset out of a family of subsets. The authors should give more emphasis on the related concept of \alpha-regret that basically competes with a multiplicative approximation to the optimal solution. This concept was introduced in previous papers and deserves some discussion, including the guarantees provided by these previous papers w.r.t. it. In general, I felt that there is more room for comparison with the previous works of combinatorial bandits. For example, what does the feedback contain in these papers? What are the guarantees? What are the differences in the methods used to solve the problem?

To conclude, the paper is overall well written and well organized. The problem dealt with consists of a new view to the combinatorial bandit problem that requires a new type of algorithm and analysis. The exact setup of the feedback lacks motivation, but overall the idea of an analysis aimed to competing with a greedy algorithm is sufficiently interesting and motivated.

Minor comments: - There are multiple grammar errors. Here are a few: o Line 62: "in offline problem" -> "in an offline problem" o Line 83: "... regret guarantee" -> "... regret guarantees" o Line 84: "need artificial" -> "need an artificial" - Line 246: there is a definition of \mu that should probably be \hat{f} - Line 285, equation 5: should there be a max{0, ___} in the definition? - Section 4.3: Moving from a known horizon to an unknown horizon via resetting the learning algorithm is a very common trick called the doubling trick (squaring when the regret is logarithmic and the interval length is squared rather than doubled). Please mention this since as written the writing indicates you invented the method - Line 355: In the inline equation, should the definition have a max with 0? - Line 364: Should S in the subscript be S_k ? - Line 408: The definition of {\cal F} seems wrong. Please make sure it captures the family of subsets you intended it to capture.
Summary: The paper is overall well written and well organized. The problem dealt with consists of a new view to the combinatorial bandit problem that requires a new type of algorithm and analysis. The exact setup of the feedback lacks motivation, but overall the idea of an analysis aimed to competing with a greedy algorithm is sufficiently interesting and motivated.

Author Feedback
Author rebuttal: We would like to thank all reviewers for their valuable comments. All our response below will be properly added into the final version of the paper.

On regret bound O(1/\Delta^2 \log T) compared with O(1 / \Delta \log T) (Reviewers 1 and 2): If we know that the problem instance has linear reward function with a matroid/polymatroid constraint, we can easily extend OG-UCB and make it behave essentially the same as OMM [23]. The key is to merge those equivalent arms. More formally, we call two arms a=e|S and a'=e|S' equivalent if the marginal rewards of both arms follow the same distribution. For these equivalent arms, we merge estimator \hat{X}(a) and \hat{X}(a') (and the counters N(a) and N(a')). This applies to the setting of linear reward function [23] and [33]: the marginal value of choosing arm e|S or e|S' only depends on element e and is irrelevant to S or S', and thus we merge all such arms e|S and e|S', such that observations of one arm refine the estimation of its equivalent arms. In this case, OG-UCB and OMM behave essentially the same (except on some minor manipulation of time counter t'). Therefore, the analysis of OMM applies to OG-UCB: OG-UCB has regret bound O(1/\Delta \log T) and the memory cost is O(n).

On the lower bound (Reviewers 1 and 2): In general, our algorithm (OG-UCB) with the above extension can achieve O(1/\Delta \log T) upper bound with a matching lower bound for the setting of linear reward function under a matroid or polymatroid decision class. While for a non-linear reward function and a wider decision class, our lower bound result shows that there could be problem instances in which \Omega(1/\Delta^2 \log T) is needed for a class of algorithms (including our algorithms in the paper) that uses layer-by-layer information only, which means the algorithm selects a decision sequence (S_0, S_1, ..., S_k) to play by individually selecting each S_i, and when selecting each S_i = S_{i-1} \cup {e_i}, only prior feedback information on S_{i-1} \cup {e} for any e is used (we will explicitly add this condition on the class of algorithms in the final version). The detailed explanation is as follows. For each step in the flow of decision sequence, the gap of a greedy arm and other non-greedy arms is \Delta. For any algorithm using layer-by-layer information, it needs to identify the greedy arm by playing any non-greedy arm for \Omega(1/\Delta^2 \log T) times in expectation by the result of Lai and Robin [25]. In our problem instance in the lower bound proof, a small estimate error (caused by the small gap \Delta) at one step can cause significant difference in the final regret if any non-greedy sequence is chosen. As a result \Delta cannot be cancelled. Thus, the lower bound is of \Omega(1/\Delta^2 \log T). Furthermore, when \Delta is set appropriately to o(1/m) (say, \Delta = 0.01 / m^1.1), then \Delta_\max = 1/2 + o(1). Therefore, we can get the \Omega(\Delta_\max / \Delta^2 \log T) lower bound.

We further remark that if we consider beyond algorithms using layer-by-layer information, a similar problem instance as in the lower bound proof can be used to show a regret lower bound with an exponential term: \Omega(exp(n, m) / \Delta \log T). The intuition is that all the first m-1 layers have essentially no reward on the decision sequence and all reward is on the last layer, which makes it similar to the classical multi-armed bandit with (n choose m) arms. Therefore the lower bound is in the order of \Omega(exp(n, m) / \Delta \log T). This means we cannot have O(poly(n,m)/\Delta \log T) regret.

On comparing with contextual bandit (Reviewer 2): In contextual bandit, each context is fixed by the environment, while in our case the previous choice S in e|S is selected by the algorithm.

On the motivation of the feedback model (Reviewer 3): In general, we think it is reasonable to assume the selected decision sequence is incrementally deployed so that reward feedback for any prefix can be observed. For example, in the influence maximization of choosing k seeds, the seeds may be applied incrementally so that we observe influence spread of every prefix set of seeds.

On the space complexity of OG (Reviewer 1): n choosing m is the worse-case space complexity, while the average space complexity is linear to the regret bound, which is still polynomial even if T is exponential in n and m.

On the current motivating example being bad and our result not solving the particular problem in the motivating example (Reviewer 1): We are not sure if the motivating example the reviewer referred to is the influence maximization example we mentioned in the introduction. If so, in our supplementary material (Section G.2 and G.3), we show that our result solves the influence maximization and other stochastic online submodular maximization problems.